



# Mapping and inventorying active rock glaciers in the Northern Tien Shan (China) using satellite SAR interferometry

Xiaowen Wang[1,2], Lin Liu[1], Lin Zhao[3], Tonghua Wu[3], Zhongqin Li[3], Guoxiang Liu[2]

[1] Earth System Science Programme, Faculty of Science, The Chinese University of Hong Kong, Hong Kong, China

[2] Department of Remote Sensing and Geospatial Information Engineering, Southwest Jiaotong University, Chengdu, China

[3] Northwest Institute of Eco-Environment and Resources, Chinese Academy of Sciences, Lanzhou, China

*Correspondence to*: Lin Liu (liulin@cuhk.edu.hk)

**Abstracts:** Rock glaciers are widespread in the high mountains of western China. However, they have not been systematically investigated for more than two decades. In this study, we propose a new method that combines SAR interferometry and optical images from Google Earth to map active rock glaciers (ARGs) in the Northern Tien Shan (NTS) in China. We compiled an inventory that includes 261 ARGs and quantitative information about their locations, geomorphic parameters, and down-slope velocities. Our inventory shows that most of the ARGs are moraine-derived (68 %) and facing north-east (56 %). The altitude distribution of ARGs in the western NTS is significantly different from those located in the eastern part. The down-slope velocities of the ARGs vary significantly in space, with a maximum of about 114 cm yr$^{-1}$ and a mean of about 37 cm yr$^{-1}$. Using the ARG locations as a proxy for the extent of alpine permafrost, our inventory suggests that the lowest altitudinal limit for the presence of permafrost in the Northern Tien Shan is about 2500–2800 m, a range determined by the lowest ARG in the entire inventory and by a statistics-based estimation. The successful application of the proposed method would facilitate an effective and robust effort to map rock glaciers over major mountain ranges and provide important datasets to improve mapping and modeling permafrost distribution in vast western China.

**Keywords**: Rock Glacier, Tien Shan, InSAR, Mountain Permafrost




# 1 Introduction

Rock glaciers are lobate or tongue-shaped landforms consisting of mixtures of unconsolidated rock debris and ice in an alpine environment. They are commonly regarded as visible expressions of permafrost presence in periglacial areas (Boeckli et al., 2012; Brenning et al., 2007; Haeberli et al., 2010; Humlum, 2000). Rock glaciers can take several thousand years to develop, so a knowledge of their distribution can provide valuable information about past incidences of permafrost and related climatological characteristics (Humlum, 1998; Konrad et al., 1999; Millar et al., 2015; Rode and Kellerer-Pirklbauer, 2012; Sorg et al., 2015).

According to their kinematic status and the presence of internal ice, rock glaciers can be classified into three groups: active, inactive, and relict. Active rock glaciers (ARGs) typically creep downslope cohesively under gravity by a few centimeters to a few meters per year (Berger et al., 2004; Kääb et al., 2003; Necsoiu et al., 2016). Inactive rock glaciers are those that contain subsurface ice but do not move due to the permafrost degradation or unfavorable topography. Relict rock glaciers do not contain subsurface ice.

Vertically distributed within a narrow band just above the lower limit of permafrost, active and inactive rock glaciers (collectively termed as "intact" rock glaciers) are indicators of alpine permafrost. Many studies have used active and inactive rock glaciers to map local-scale indicators of permafrost occurrence in major mountain ranges such as the European Alps, the Rocky Mountains, the Andes, and the Hindu-Kush-Himalayan region (e.g., Barsch 1978 ; Brenning and Trombotto, 2006; Janke 2005, Boeckli et al., 2012; Bollmann et al., 2015; Schmid et al., 2015). However, it is difficult to assess the kinematic status of rock glaciers directly (either from the field or remote sensing) and therefore challenging to distinguish active and inactive rock glaciers (Onaca et al., 2016; Sattler et al., 2016). If one only uses ARGs, it would be more accurate and robust to infer the permafrost distribution (Sattler et al., 2016).

It is frequent to use geomorphological features to identify ARGs. Rock glacier movements commonly manifest in viscous flow morphologies on the ground surface, such as the presence and development of ridges, furrows, and steep fronts (Berthling, 2011; Clark et al., 1998; Ikeda and Matsuoka, 2006; Janke et al., 2015). Based on the indications of these geomorphological features, one can map ARGs by visual interpretation through field surveying or aerial/satellite images (Brenning et al., 2012; Esper Angillieri,




2009; Lilleøren et al., 2013; Schmid et al., 2015; Scotti et al., 2013). However, field surveys are usually scarce because most rock glaciers are located in the remote areas and are difficult to access. Because optical images are often affected by clouds and shadows, it is not always feasible to use them to distinguish ARGs
from inactive and relict rock glaciers.

In addition to visually interpreting geomorphological features, measuring surface displacements can also be used to identify active rock glaciers. Various terrestrial geodetic techniques such as differential GPS and laser scanning have been used (Avian et al., 2009; Bollmann et al., 2015; Lambiel and Delaloye, 2004; Wirz et al., 2015). However, these terrestrial techniques are expensive and only offer limited spatial coverage.
They have therefore mostly been used in detailed case studies.

In recent years, satellite synthetic aperture radar interferometry (or InSAR) has emerged as a unique technique to investigate the dynamics of rock glaciers (Barboux et al., 2015; Kenyi and Kaufmann, 2003; Lilleøren et al., 2013; Liu et al., 2013; Necsoiu et al., 2016; Strozzi et al., 2010). Compared with optical remote sensing techniques, InSAR provides an all-weather imaging capability for measuring ground surface
deformation with centimeter to sub-centimeter precision. Thus, the InSAR measurements can be used to determine the kinematic status of rock glaciers (especially their velocities, the upper and lower boundaries), which complement the limitations of optical images. However, the combined use of InSAR measurements and optical images for mapping ARGs in a vast, regional area is still rare. In this study, we proposed an integrative method to identify and map ARGs with improved accuracy, effectiveness, and efficiency by (1)
detecting moving landforms using InSAR and (2) interpreting geomorphic features from images in Google Earth.

Despite that rock glaciers are abundant in the high mountains of western China, they have not been systematically investigated. A small number of studies conducted more than two decades ago examined the surface lithology and geomorphologic features of rock glaciers at several mountains including Tien Shan,
Kunlun Shan, and Hengduan Shan (Cui and Zhu, 1989; Li and Yao, 1987; Zhu, 1992a; Zhu et al., 1992b; Liu et al., 1995). These studies mainly focused on the shapes, classification, and topo-climatic conditions of rock glaciers based on the interpretation of aerial photos and field surveys. There is a lack of studies on surface velocities of rock glaciers or an inventory containing the locations of the surveyed rock glaciers.



Additionally, due to the limited number of rock glaciers investigated in the field surveys, there has not been
any discussion on the permafrost distribution for these high mountains based on locations of rock glaciers.

In this study, we applied the new integrative method to identify and map the ARGs in the Northern Tien
Shan (NTS) region as an example study area. We determined the geographical locations of ARGs based on
the proposed method. Then, we systematically summarized their geomorphic and climatic attributes, as well
as their surface velocities. Finally, we inferred the lower limit of permafrost in the NTS from the inventoried
ARGs, offering a modern, comprehensive, and rigorous update from the previous field-based estimates by
Qiu et al. (1983).

## 2 Study area

The Tien Shan is a major mountain system in Central Asia, with a mean altitude of more than 4000 m above
sea level. The portion of Tien Shan mountain system in China almost divides Xinjiang into two large basins,
the Tarim Basin in the south and the Junggar Basin in the north, with the Gurbantunggut and the
Taklamakan Deserts lying in hinterland basins (Fig.1). Due to its geographical location and rugged
topography, Tien Shan possesses the ideal climatic conditions for the creation and preservation of mountain
permafrost. Qiu et al. (1983) estimated that permafrost in the Chinese part of the Tien Shan range covers
approximately $6.3 \times 10^4$ km$^2$. In this study, we focus on the Northern Tien Shan (NTS) region in China (82 °–
87 °E, 42 °–44 °N), a large branch of the Tien Shan striking N20 °W from northwest to southeast (Fig. 1).
The eastern and western parts of the NTS divide along longitude 86 °E, and possess distinct climatic
characteristics (Zhu et al., 1992b). The west is typically wet and cold. In contrast, the east is dry and cold.
The east-west contrast is also evident in the number and size of mountain glaciers. The west has more and
larger glaciers than the east.

Previous studies of rock glaciers in the NTS are limited with most of them concentrating in the Urumqi
River source region. Interpreting aerial photos acquired in the late 1980s and early 1990s, several
researchers (Cui and Zhu, 1989; Zhu et al., 1992a, 1992b; Liu et al., 1995) identified hundreds of rock
glaciers in the NTS. Cui and Zhu (1989) and Zhu et al. (1992b) measured the position changes of marked
boulders on some rock glaciers and estimated the surface velocities. They found that most of the identified
rock glaciers are tongue-shaped, located at altitudes between 3300 m and 3900 m, and facing north. The



rock glaciers in the western part of the NTS are generally larger and faster than those in the eastern part. They reported maximum surface velocities of 193 cm yr$^{-1}$ and 75 cm yr$^{-1}$ in the western and eastern parts, respectively. However, previous studies did not provide any detailed information such as geographical and geomorphic attributes of the identified rock glaciers.

# 3 Methodology

In this section, we first present the details of the InSAR processing for detecting the moving targets in the interferogram and the rules used to identify ARGs in Google Earth by visually interpreting their geomorphic features (Section 3.1). We then describe the method to derive the down-slope velocities of the ARGs (Section 3.2). Finally, we list and explain the criteria for compiling an inventory that contains the geomorphic and dynamic attributes of the identified ARGs, followed by a description of the uncertainty analysis (Section 3.3).

## 3.1 Identifying and mapping active rock glaciers

### 3.1.1 Identifying ground movements using SAR interferometry

The InSAR technique detects the ground movements by exploiting the phase difference between SAR images acquired at different times (Rosen et al., 2000). The phases in the differential interferogram are wrapped within $-\pi$ to $\pi$. One phase cycle corresponds to one-half wavelength (e.g., 11.8 cm for L-band) of ground displacements along the line-of-sight direction (Rosen et al., 2000; Wang et al., 2015). We used wrapped interferograms rather than unwrapped displacement maps to detect ARGs because wrapped phases in the interferogram show apparent variations visually even if the ground movements are small, for instance, at centimeter level.

We used seven orbital paths of L-band ALOS PALSAR images to cover the entire study area. All images were collected at a radar off-nadir angle of 38.7 °in HH polarization and have a pixel spacing of 4.7 m in slant range and 3.2 m in azimuth. For each orbital path, we used one image pair to form one interferogram. To achieve high interferometric coherence, we selected the images pairs with temporal spans of either 46 or 92 days and critical perpendicular baselines smaller than 600 m for further processing. Table 1 supplies detailed information on the interferometric pairs. With the availability of PALSAR images, we implemented





the interferometric processing by using GAMMA software (Wegmüller et al. 1998). The topographic phases were estimated and removed by using the 1-arc-second SRTM digital elevation model data with a spatial resolution of about 30 m. We applied multi-looking operation (2 looks in range and 5 looks in azimuth) and
filtered interferometric phase using the adaptive Goldstein filter with a window size of 8 by 8 pixels.

However, we were unable to map the ARGs solely based on the phase variations in the interferograms. First, the phase variations can be caused by several kinds of error sources in SAR interferograms such as residual topography contribution, de-correlation noise, and atmospheric delay (Liu et al., 2013; Strozzi et al., 2010). Second, the moving targets may correspond to other periglacial landforms such as debris flow, solifluction,
and protalus lobes (Scotti et al., 2013). Therefore, we decided to overlay the interferograms on optical satellite images in Google Earth to identify the ARGs by visually interpreting their morphological features.

### 3.1.2 Identifying active rock glaciers using Google Earth images

ARGs are characterized by distinct flow features and structural patterns. Transversal or longitudinal flow features (ridges and furrows) are common on ARGs as a consequence of the deformation of internal ice
(Berthling, 2011; Clark et al., 1998; Haeberli et al., 2006; Humlum, 2000). Many ARGs also have structural patterns such as the steep frontal slopes and side slopes with swollen bodies. Due to the constant supply of talus or debris, the surface textures of ARGs are usually different from the surrounding slopes, and their surface slopes have little or no vegetation. Based on these criteria, we visually examined the landforms in the Google Earth image that correspond to the moving targets in the interferograms and identified the ARGs.
We distinguished ARGs from debris-covered glaciers based on their different visual features and rooting zones on optical images. Debris-covered glaciers are usually covered with uniformly thin debris layer, whereas rock glaciers' debris cover is less homogenous and coarser. The rooting zones of debris-covered glaciers are continuous with clean glacier ice (Davies et al., 2013; Lukas et al., 2007). We marked the geographical location of each identified ARG and delineated its outline using Google Earth.

Based on the sources of rock materials that input into the surfaces and internal bodies of rock glaciers, we further classified the inventoried ARGs into two categories: moraine-derived active rock glaciers (MARGs) and talus-derived active rock glaciers (TARGs). MARGs develop beneath the end moraines of small glaciers and transports mainly reworked glaciogenic materials (Lilleøren et al., 2013; Scotti et al., 2013). TARGs develop below talus slopes in which there is no visible ice upslope of the landform body, and hence transport



mainly rock fragments generated from the adjacent rock walls (Lilleøren and Etzelmüller, 2011). For

example, Figs. 2a and 2b show a typical MARG and TARG, respectively.

We demonstrate the method described above using the identified ARGs in the zone "Z" (see its location in

Fig. 1) which has an area of about 90 km$^2$. Fig. 3a shows the wrapped interferometric phase for this zone.

The interferogram was made using images taken 46 days apart in 2007, on September 2 and October 18,

respectively. This interferogram has good coherence with a mean value of 0.82, which makes it easy to

identify the moving targets based on the phase variations. In the zone "Z", we found a total number of 10

moving targets. We then visually checked the surface geomorphic features of each of the moving targets on

the Google Earth images (Fig. 3b) and identified 7 of 10 as ARGs. For the other 3 moving targets, two are

artifacts due to phase errors, and the other one is an alluvial fan other than rock glaciers, we grouped all

these 3 as none rock glaciers.

### 3.2 Deriving the surface velocity

We further estimated ground displacements using the InSAR measurements. We unwrapped the

interferograms using the minimum cost flow method by selecting a reference point in the flat area with high

coherence. We chose a local reference patch (3×3 pixels) near but outside each ARG to calibrate the

unwrapped phase. We then calculated the surface velocities of the ARGs by dividing the displacements by

the temporal span of the interferogram. We then converted the InSAR-estimated velocities in line-of-sight

direction ($V_{LOS}$) into down-slope direction ($V_{slp}$) using the following equation (Liu et al., 2013):

$$V_{slp} = \frac{V_{LOS}}{\sin(\alpha-\beta)\sin\theta_{inc}cos\theta_{slp}+cos\theta_{inc}sin\theta_{slp}} \tag{1}$$

where $\alpha$ is the flight direction of the SAR satellite measured from the north; $\theta_{inc}$ is the local incidence

angle which can be calculated based on the SAR looking geometry and local topography (i.e., digital

elevation model) data; $\beta$ and $\theta_{slp}$ are the aspect angle and slope angle of the ARG, respectively. Finally,

we calculated the mean value of all pixels within each ARG to represent its overall velocity.

### 3.3 Characterizing the geomorphic and climatic parameters of active rock glaciers

To characterize the geomorphic features of the ARGs, we first determined the initial line point (ILP) and

front line point (FLP) of each ARG, respectively. According to the definition given by Humlum (1998) and





Sattler et al. (2016), the ILP is usually located in the rooting zone of a rock glacier and represents the place where permafrost creep starts. We determined the initial line at the rooting zone of an ARG as where the InSAR phases start to vary from the regional background. If InSAR coherence was low in the rooting zone, we used morphologic features such as ridges and furrows instead to determine the initial line. We then used the central point of the initial line as the ILP for an ARG. The FLP, the lowest place where a rock glacier can reach, could be easily identified from InSAR and Google Earth images. Fig. 4 gives an example showing the determined ILP and FLP for a typical TARG.

We also quantified the geomorphic (including altitude, area, length, shape, slope angle, and aspect angle) and climatic parameters (Mean Annual Air Temperature, MAAT and potential income of solar radiation, PISR) of each ARG. We determined the altitudes of the initiation line point and the front line point from the interpolated SRTM digital elevation model. We then calculated the slope distance ($L$), mean slope angle ($\theta_{slp}$) and the mean aspect angle ($\beta$) based on the coordinates of ILP and FLP. Dividing the area by the square of length gives the aspect ratio, which distinguishes tongue-shaped (aspect ratio <1) from lobate-shaped (aspect ratio > 1) ARGs. We extracted the MAAT at the FLP from the NCAR-NCEP global climate reanalysis and interpolated station measurements released by Gruber et al. (2012). We calculated the PISR over one year (unit: W/m$^2$) using the method described by Kumar et al. (1997) with the assistance of the digital elevation model.

With some assumptions, we performed a simple error analysis about the rock glacier parameters we estimated. First, with the high-resolution InSAR phase maps and Google Earth images, we estimated that the location of each ARG and FLP/ILP we determined is accurate within 50 m (i.e., three times of the InSAR spatial resolution). The uncertainties of the lower limit of the permafrost distribution are determined by the errors of the SRTM digital elevation model, which has a nominal vertical accuracy of less than 16 m (Farr et al., 2004). The uncertainties of the other geometric parameters can be estimated by error propagation. For instance, given a rock glacier with a length of 1000 m, an aspect angle of 80 °, and a slope angle of 12 °, we estimated that the uncertainties for length, aspect and slope are about 67.2 m, 3.8 °, and 1.6 ° respectively. We calculated that the mean coherence of each ARG and consequently estimated the uncertainty of the mean InSAR measurements based on the empirical method of Hanssen (2001). We then estimated the




uncertainties of surface velocities by using the Equation (1). The uncertainties of other geometric parameters (i.e., area and aspect ratio) are difficult to quantify due to the irregular shapes of the ARGs.

215 Finally, we compiled a spreadsheet to summarize the characteristics of the inventoried ARGs (see the supplementary files). Our inventory lists the geographical locations (including longitude, latitude, and altitude), the geomorphic attributes (including area, length, aspect ratio, slope angle, and aspect angle), the climatic factors (i.e., PISR and MAAT), and the surface velocity and its associated uncertainty of each rock glacier.

# 4 Results

We identified and mapped a total of 261 ARGs across the NTS (see Fig.1). Table 1 lists the number of ARGs identified in each satellite path. For any ARG located in the overlapping regions of two adjacent paths, we selected the interferogram that had a higher coherence on the ARG to derive the surface velocity. In the following subsections, we first present a general statistics of the inventory (Section 4.1). We then 225 characterize the altitude distribution of the ARGs in Section 4.2 and describe of the surface velocities of the ARGs in Section 4.3.

## 4.1 General statistics of the inventory

Table 2 provides the overall information of the inventoried ARGs in the NTS. Among the 261 ARGs we mapped, 177 (68 %) are MARGs and 84 (32 %) are TARGs. The MARGs are generally longer, steeper and 230 larger than the TARGs. The mean FLP altitudes for three groups, namely, all the ARGs, the MARGs, and the TARGs, are 3175 m, 3209 m, and 3101 m, respectively. Their mean ILP altitudes are 3486 m, 3515 m, and 3425m, respectively. These figures indicate that the favorable altitude for ARGs in the NTS is above 3000 m, which is consistent with the previous reports of Liu et al., (1995) and Zhu et al., (1992a). Most of the ARGs are tongue-shaped, with a mean aspect ratio of about 0.25. The mean area of the MARGs is nearly 235 twice as large as that of the TARGs. The ARGs occupy a total area of about 91.63 km$^2$, 83 % of which is occupied by MARGs.

Taking all ARGs as a single group, their ILP altitudes and FLP altitudes both show nearly norm distribution (Figs. 5a and 5d). The majority of ARGs have the ILP altitudes higher than about 3300 m (Fig. 5a) and the





FLP altitudes higher than about 3000 m (Fig. 5d). IPLAs and FLP altitudes show different statistic

distributions between the MARGs and TARGs. The histogram of ILP altitude for the MARGs shows a sharp

increase at an altitude of about 3300 m and a sharp decrease at about 3800 m (Fig. 5b). In contrast, the

histogram of ILP altitude for the TARGs shows a nearly even distribution, though with some degree of

randomness (Fig. 5c). The FLP altitudes for MARGs and TARGs both show a nearly norm distribution

(Figs. 5e–f). The minimum, median, and maximum FLP altitudes of the MARGs are all higher than those of

the TARGs, indicating that the MARGs develop on higher slopes than the TARGs.

In the inventory, 70 % of the ARGs (number: 182) are located on north-facing (defined between NW and

NE) slopes (Fig. 6a), with the NE aspect predominating (56 %), indicating the north-facing slopes are the

most suitable slopes for rock glacier development and formation in the NTS. The number of north-facing

MARGs and TARGs are 133 and 49, respectively, accounting for the 75 % and 58 % of the total numbers of

MARGs and TARGs, respectively. The difference is probably because the formation of MARGs is

controlled by the evolution of glaciers in the source regions. We also noted that the majority of glaciers in

the NTS are north-facing, as controlled by topographic and geometric factors (Kaldybayev et al., 2016; Li

and Li, 2014). Compared with the MARGs, the TARGs are more evenly distributed across all aspects. This

pattern is consistent with that found in rock glaciers in the central Italian Alps and Southern Carpathians

(Onaca et al., 2016; Scotti et al., 2013).

The subsurface thermal condition is another major factor controlling the development of ARGs, which is

influenced by solar radiation. Fig. 6b shows the variations of PISR with the aspect angles. The mean value

of PISR for the north-facing and south-facing slopes are $8.83 \times 10^5$ W/m$^2$ and $10.46 \times 10^5$ W/m$^2$, respectively.

The south-facing ARGs typically receive higher PISR than their north-facing counterparts. As most of the

ARGs are north-facing, we conclude that slopes with lower PISR favor the presence of ARGs. This is

probably because lower PISR helps to preserve ground ice inside ARGs.

### 4.2 Altitudinal distribution of the active rock glaciers

To explore how the factors would impact the altitude distribution of the ARGs, we calculated the Pearson

correlation coefficients between the FLP altitudes and the environmental factors listed in Section 3.3. FLP

altitude denotes the lowest altitude that an ARG can reach and is frequently used as the lower boundary for

the presence of mountain permafrost locally (Lilleøren et al., 2013; Schmid et al., 2015). For the ARGs in



our inventory, the correlation coefficients between the FLP altitudes and the longitude, latitude, MAAT, PISR, slope angle, and aspect angle are 0.47, -0.52, -0.86, 0.29, -0.35, and -0.08, respectively.

The simple correlation analysis confirms the strong influence of the geographical locations on the altitude distribution of the ARGs (Fig. 7a and 7b). The FLP altitudes generally increase from west to east and decrease from south to north. This spatial pattern is mainly because the NTS lays with a strike of about N20°W extending from northwest to southeast. Along this direction, the air temperature increases with lower latitude, thus increasing the lowest altitude for the development of rock glaciers.

Additionally, we found different altitude distributions of ARGs in the western and eastern parts of the NTS (Fig. 7a). To test the statistical significance of the two sub-regions, we performed a T-test to the FLP altitudes of the ARGs. The P-value of T-test is smaller than 0.01 at the 95 % confidence level. Therefore, we conclude that the altitude distribution in the western NTS is significantly different from those located in the eastern part. This difference probably results from the distinct climate backgrounds between the western and eastern parts of the NTS as described in Section 2.

The correlation analysis also reveals the relative importance of various factors to the rock glacier velocities. The highest correlation coefficient is found at the factor "MAAT", implying that the MAAT may influence ground thermal conditions, thus governing the formation, evolution, and survival of the ARGs. A moderate positive correlation is observed at the factor "PISR". Receiving a higher PISR, the ground temperature is likely higher and the internal ice is more likely to melt. The correlation between the slope angles and FLP altitudes are associated with the favorable presence of TARGs on the steep slopes. We found no obvious correlation between the aspect angles and FLP altitudes.

### 4.3 Surface velocities of the active rock glaciers

In this section, we present the surface velocities of the inventoried ARGs measured from InSAR. We were conservative in conducting statistical analyses on the surface velocities by only including the ARGs with a mean coherence higher than 0.3 and a mean velocity larger than 5 cm yr$^{-1}$. Finally, the surface velocities of 170 ARGs (comprising 110 MARGs and 60 TARGs) were documented (the bottom figure of Fig. 8). The surface velocities of the 170 ARGs show a spatially heterogeneous pattern over the NTS. Most of ARGs (84 %) have surface velocities lower than 60 cm yr$^{-1}$. The maximum surface velocity is 114 cm yr$^{-1}$. The



mean velocity and the standard deviation of all ARGs are 37 cm yr$^{-1}$ and 21 cm yr$^{-1}$, respectively. The

maximum velocity uncertainty is 28.8 cm yr$^{-1}$, and the mean uncertainty is 7.5 cm yr$^{-1}$.

We selected two pairs of ARGs to inspect the spatial variations of surface velocities closely (Figs. 8a–b).
The two ARGs in each pair are adjacent to each other. Therefore, we assumed that they developed in similar
topo-climatic environments (e.g., similar altitude, MAAT, and precipitation). The surface velocities of
'ARG1' show a spatial homogeneity pattern while these of the other three all show a spatially heterogeneous

pattern. The fastest moving zones are located near the center, the rooting zone, and the front of 'ARG2',
'ARG3', and 'ARG4', respectively. This significant heterogeneity of surface velocities within ARGs
indicates that the kinematics of surface movements are complex. Previous studies have demonstrated that the
surface velocities ARGs are not only influenced by topo-climatic factors but also the ice condition and
internal structure of ARGs (Haeberli et al., 2006; Schoeneich et al., 2015; Serrano et al., 2010). However, it

is difficult to determine and quantify the ground ice conditions or ground thermal conditions of the ARGs in
this study. Nevertheless, the spatial variabilities of velocities at both the local and regional scales suggest
that the dynamics of ARGs are not controlled by any single factor, but a combination of many factors.

**4.4 Lower limit of permafrost occurrence in the Northern Tien Shan**

We used the inventoried ARGs as a proxy of permafrost to infer lower limit of permafrost occurrence in the

NTS. The previous estimates of the lower limit of permafrost presence in the NTS were mainly based on the
field surveys conducted by Qiu et al. (1983). The field surveys they conducted are located near the Urumqi
River source region (87.00 °E, 43.07 °N) in the eastern of NTS and near the Haxilegen Pass (84.45 °E,
43.77 °N) region in the west, respectively. However, these site-based estimations may not be representative
of the whole study area. Therefore, the ARGs we inventoried are valuable for re-estimating the permafrost

distribution for this region. Permafrost is unlikely to develop below the FLP, and the proportion and extent
of permafrost usually increase towards higher altitudes. Because our inventory shows that MARGs generally
develop on higher slopes than TARGs (see Section 4.1), here we only used the FLP altitudes of TARGs to
estimate the lower limit of permafrost.

We grouped the inventoried TARGs into different categories based on their locations (i.e., east and west of 86 °E)

and aspect angles (i.e., north-facing and south-facing) to infer the lower limit of permafrost under different
environmental conditions. The first row of Table 3 summarizes their altitudinal limits: 2490 m and 2583 m for



the north-facing and south-facing slopes in the west, respectively; and 2895 m and 3088 m for the north-facing and south-facing slopes in the east, respectively. Overall, the altitudes of the inventoried ARGs increase from west to east as shown in Fig. 7a.

However, these estimates may not give an entirely accurate picture of the permafrost limit for the whole study area, because they are based on a few specific ARGs that have the lowest FLP altitudes in the inventory. For example, the TARG with the lowest FLP altitude (i.e., 2490 m) is located on a north-facing slope in the western part of the NTS (shown in Fig. 9c, outlined by the blue polygon). We cannot rule out the possibility that this particular ARG may be a local anomaly and therefore cannot fully represent the permafrost presence in the

broader NTS. Thus, we adopted an alternative method in a statistical sense, which approximates the lower limit of permafrost by subtracting the standard deviation of all FLP altitudes in the inventory from their mean values. The second row of Table 3 lists these statistics-based estimates: 2775 m and 2831 m for the north-facing and south-facing slopes in the west, respectively; and 2998 m and 3038 m for the north-facing and south-facing slopes in the east, respectively. The largest difference between the statistics-based and lowest-altitude-based is

285 m, indicating that both estimates are fairly consistent, although the statistics-based values are normally higher. Therefore, we suggest the lower limit for the permafrost distribution in the NTS is about 2500–2800 m.

   The third row of Table 3 summarizes the field-based estimates of the lowest altitude of permafrost given by Qiu et al. (1983). We note that the results of the field surveys are all higher than these inferred from our lowest-altitude-based estimates. We also note that the field-based results are all lower (and higher) than our

statistics-based estimates for the north-facing (and south-facing) slopes. The largest differences between the lowest-altitude-based and the field-based estimates are 467 m and 162 m for the western and eastern parts, respectively. The largest discrepancies between the statistics-based and field-based estimates are 219 m and 212 m, respectively. One possible reason for these discrepancies is that our results based on ARG inventory give regional estimates, while the field surveys give site-specific estimates. Additionally, the ARGs move

downslope and the fronts of those low-lying ARGs may reach places free of permafrost.

# 5 Discussion

## 5.1 Limitations of using active rock glaciers to determine the lower limit of permafrost



The moderate discrepancies between our estimates and the field surveys of the altitudinal limits for permafrost occurrence in the NTS demonstrated the feasibility of our method to infer the permafrost

distribution. However, several factors still exist and limit the accuracy of our method for quantifying the lower limit of permafrost. These factors include (1) the ambiguity about the genesis of rock glaciers, (2) the high heterogeneity of ground thermal regime, and (3) the uncertainties in the use of FLP altitudes of the ARGs to infer permafrost limit.

First, it is still debatable to use active rock glaciers as representatives of the exact permafrost boundaries.

There are two contradictory viewpoints regarding the genesis of rock glaciers. One holds rock glaciers are permafrost creep features and form in a permafrost environment. The other one holds that rock glaciers form through a continuum of glacial to periglacial processes and thus they may also be related to glaciers in non-permafrost environments (Berthling, 2011; Clark et al., 1998). To avoid this ambiguity, we only used the talus-derived ARGs in this study to infer the lower limit of permafrost in the NTS.

Second, as shown in Section 4.2, the altitudinal distributions of the ARGs in the NTS indicate that the presence of rock glaciers is primarily governed by the ground thermal regime. However, the ground thermal regime in the alpine region can be highly heterogeneous in space. Thus, it has to be clear that a specified rock glacier is just the local-scale indicator of permafrost occurrence. This is why we estimated the lower limit of permafrost occurrence in two, i.e., lowest-altitude-based and statistics-based, perspectives.

Lastly, several sources of uncertainty exist in the use of FLP altitudes of ARGs to infer the lowest altitude of permafrost occurrence in the NTS. On one hand, ARGs move downslope and may export permafrost into lower altitudes that are climatically unfavorable for permafrost formation. Thus, our estimates may give a bias towards a lower than the true permafrost limit. On the other hand, we did not identify inactive rock glaciers, which are also indicators of permafrost occurrence. Since inactive rock glaciers generally develop

at a lower altitudes than active ones (Sollid and Sørbel, 1992), our estimates may bias the permafrost limit towards a higher value. Additionally, the number of ARGs we compiled is a conservative estimation of all the ARGs in the NTS. The conservative estimation is due to two reasons. The first reason is that we may have missed some ARGs that are located inside the de-correlated regions of the interferograms or the radar shadow/layover zones. The second reason is that ARGs facing nearly north-south direction might also have

been missed, as InSAR (using images from all space missions, including ALOS) is not sensitive to ground





motions in these particular directions (Wang et al., 2015). We cannot rule the possibility that some

extremely low-lying ARGs are not included, which may lower our estimated altitude limits. These three

kinds of uncertainties justify the expression of a range of estimations rather than exact values of the altitude

limits as given in Section 4.4.

**5.2 Comparison with the existing permafrost distribution maps**

We cross-validated the inferred permafrost distribution by comparing our inventoried ARGs with the

existing permafrost distribution maps. We selected three permafrost maps that covers the NTS region: (1)

the Map of Snow, Ice and Frozen Ground in China released by Cold and Arid Regions Environmental and

Engineering Research Institute, Chinese Academy of Sciences (Shi and Mi, 1988), which we refer to as the

CAS map,   (2) the Circum-Arctic Map of Permafrost and Ground Ice Conditions released by the

International Permafrost Association (Brown et al., 1998), referred to as the IPA map, and (3) the global

permafrost zonation Index (PZI) map developed by Gruber et al. (2012). The CAS map is compiled based on

the field observations, aerial photographs and terrain analysis with a scale of 1:4000000. The frozen ground

in the CAS map is classified into high-latitude, plateau, and mountain permafrost. The IPA map adopts the

classes of continuous, discontinuous, sporadic, and isolated patches of permafrost extent with a scale of

1:1000000. The PZI map is generated based on the relationships between permafrost extent and long-term

MAAT data with a ground resolution of 1 km. The PZI values range from 0 to 1. Values close to 1 represent

permafrost existing in nearly all conditions, while values close to 0 represent permafrost presenting only in

the most favorable conditions. The PZI uncertainty is expressed with "fringe of uncertainty", referring to a

conservative or an anti-conservative variant than the "normal" scenario that considers the possibility of

<10 % permafrost area as well (Gruber, 2012). Schmid et al. (2015) assessed the qualities of IPA map and

the PZI map by comparing them with 702 rock glaciers identified from Google Earth images in the vast

Hindu Kush Himalayan region with a continental climate. They found that the PZI map has a better

agreement with the distribution of rock glaciers than the IPA map. However, Sattler et al. (2016) argued that

the PZI may underestimate of permafrost extent in maritime climate setting by comparing the PZI map with

modeled permafrost distribution based on inventoried rock glaciers in Southern Alps, New Zealand.

Fig. 9 shows the comparison of the ARGs we inventoried in the NTS with the three permafrost maps

described above. The dashed white and yellow lines in Fig. 9a outline the mountain permafrost extent from



the CAS map and the discontinuous permafrost from the IPA map, respectively. The background color map

in Fig. 9a shows the PZI values. We found that 256 and 249 ARGs are located within the permafrost extents

of the CAS and IPA maps, respectively, indicating that the distribution of the ARGs we inventoried are in

good agreement with these two permafrost maps. We also found that most of the ARGs we inventoried are

located in areas with high PZI values. As the PZI map has the highest spatial resolution among these three

permafrost maps, we further compared our inventoried ARGs with the PZI map.

We randomly selected two sub-regions in the western (Fig. 9b) and eastern (Fig. 9d) of the NTS,

respectively, to examine the distributions of ARGs in the PZI map. It is evident that most of the MARGs are

located above the PZI contour line of 0.8. We select the zone shown in Fig. 9c as another example as it

contains both the MARG and TARG that have the lowest FLP altitude in their corresponding category. The

MARG developing in the narrow valley (location: 84.2957 °E, 44.0021 °N, outlined by the red polygon in

Fig. 9c) has an FLP altitude of 2575 m. The TARG (location: 84.2883 °E, 44.0228 °N, outlined by the blue

polygon in Fig. 9c) has an FLP altitude of 2490 m. The PZI values at the front points of these two ARGs are

still higher than 0.2.

The PZI value at the FLP of an ARG shows how favorable the lowest altitude that the rock glacier reaches

would for permafrost to exist. The statistical distribution of PZI at FLPs for all the ARGs shows that most

ARGs (73 %) we inventoried are located in the PZI interval of 0.5 to 0.9 (Fig. 10a). The PZI density peaks at

about 0.8 for the MARGs and 0.5 for the TARGs (Fig. 10b–c). The PZI density decreases towards the lower

PZI values for the MARGs, while decreases towards both ends of the distribution (Fig. 10c) for the TARGs.

This difference in PZI distribution is probably because the TARGs are located at relatively lower altitudes

than the MARGs. The PZI values at lower altitudes show a larger variability as the air temperature increases.

The mean PZI values at the FLPs are 0.64, 0.67, and 0.57 for all ARGs, MARGs, and TARGs, respectively,

indicating the spatial distribution of the ARGs we inventoried agrees well with the PZI map.

## 6 Conclusions

In this study, we successfully mapped and inventoried 261 ARGs in the NTS by combining the use of the

SAR interferometry and optical images from Google Earth. Based on the creep boundaries of the ARGs



determined by this novel method, we estimated the lower limit of the permafrost presence in the NTS. From the inventory and analyses, we draw the following conclusions:

(1) Most of the ARGs in the NTS are moraine-derived, tongue-shaped, and northeast-facing. The moraine-derived ARGs are generally longer, steeper, and larger than the talus-derived ARGs. The total area of the ARGs is 91.63 km$^2$ and moraine-derived ARGs occupy most of this area (83 %).

(2) The latitude/longitude locations have a strong influence on the altitude distribution the ARGs. We found the altitudes of ARGs generally increase from west to east and decrease from south to north. We also found distinct altitude distributions of ARGs in the western and eastern parts of the NTS (divided by 86 °E), probably due to the influence of distinct climates of these two sub-regions.

(3) We found that the lower limit of permafrost in the eastern part of the NTS is higher than in the western
part. The lower altitudinal limit for the permafrost distribution throughout the NTS is about 2500–2800 m.

(4) The distributions of the inventoried ARGs agree well with the existing CAS, IPA and PZI permafrost maps. The mean PZI values at the FLPs are 0.64, 0.67, and 0.57 for all ARG, MARGs, and TARGs, respectively.

The study presented here provides the first comprehensive and modern documentation of the characteristics of the ARGs in the NTS. It updates the lower limit of permafrost occurrence for this vast area. This inventory offers a baseline dataset for the further investigations on permafrost modeling, slope stability, and water resource, etc. The successful usage of the proposed method in the NTS implies that this method can be applied to the other high mountains in vast western China, thus closing the significant gaps in our
knowledge of rock glaciers there. This new knowledge will be useful to map and model alpine permafrost distributions in western China.

## Data availability

The locations and characteristic parameters of the 261 inventoried active rock glaciers for downloading and visualizing are available in the supplementary files. Both the KMZ file and ERIS shapefiles are provided.





## List of Acronyms


ALOS: Advanced Land Observing Satellite

ARG: Active Rock Glacier

CAS: Chinese Academy of Sciences

FLP: Front Line Point

GPS: Global Positioning System

ILP: Initial Line Point

InSAR: Synthetic Aperture Radar Interferometry

IPA: International Permafrost Association

MAAT: Mean Annual Air Temperature

MARG: Moraine-derived Active Rock Glacier

NTS: Northern Tien Shan

PALSAR: The Phased Array type L-band Synthetic Aperture Radar

PISR: Potential Income of Solar Radiation

PZI: Permafrost Zonation Index

SRTM: Shuttle Radar Topography Mission

TARG: Talus-derived Active Rock Glacier

## Acknowledgements

We thank Japan Aerospace Exploration Agency for providing the ALOS PALSAR data. We also thank the
USGS for providing the 30-meter resolution SRTM digital elevation model data. The CAS permafrost map
is provided by Environmental and Ecological Science Data Center for West China, National Natural Science
Foundation of China (http://westdc.westgis.ac.cn). The IPA permafrost map is provided by the Frozen
Ground Data Center at the National Snow and Ice Data Center (NSIDC), Boulder, Colorado, USA
(http://nsidc.org/data/GGD318/versions/2). The MAAT and permafrost map zonation data are downloaded
from http://www.geo.uzh.ch/microsite/cryodata/pf_global/. Some figures in this paper were plotted using the





Generic Mapping Tools (Wessel et al., 2013). Work at the Chinese University of Hong Kong was supported

by Hong Kong Research Grants Council Grant CUHK24300414.

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





**Table 1**. List of interferograms made from the seven paths of ALOS PALSAR data. Names of interferograms are the conjunction of acquisition dates of the two images. The last column gives the number of identified active rock glaciers (ARGs) in each interferogram.

| Path/Frame | Interferogram | Time span (days) | Perpendicular baselines (m) | Number of identified ARGs |
|---|---|---|---|---|
| 501/850 | 20070902-20071018 | 46 | 314 | 55 |
| 502/850 | 20070619-20070919 | 92 | 365 | 20 |
| 503/860 | 20070706-20071006 | 92 | 340 | 3 |
| 504/860 | 20080424-20080609 | 46 | -12 | 36 |
| 505/860 | 20080209-20080326 | 46 | 211 | 57 |
| 506/870 | 20070711-20070826 | 46 | 256 | 78 |
| 507/870 | 20090802-20090917 | 46 | 586 | 51 |

**Table 2**. Statistical summary of the inventoried active rock glaciers. Each column gives the mean values and the standard deviations (the values in brackets) of the parameters.

| | Number | FLP altitude (m) | ILP altitude (m) | Length (m) | Aspect Ratio (-) | Slope (°) | Area (km²) | Total area (km²) |
|---|---|---|---|---|---|---|---|---|
| All ARGs | 261 | 3175 (260) | 3486 (208) | 1335 (892) | 0.25 (0.19) | 14 (6) | 0.35 (0.33) | 91.63 |
| MARGs | 177 | 3209 (266) | 3515 (182) | 1519 (998) | 0.24 (0.14) | 11 (4) | 0.43 (0.36) | 76.50 |
| TARGs | 84 | 3101 (240) | 3425 (246) | 970 (485) | 0.25 (0.18) | 19 (4) | 0.18 (0.15) | 15.13 |





**Table 3**. The lower altitudinal limits of permafrost distribution in the NTS inferred from this study using the inventoried TARGs. 'This study (Min)' denotes the altitudinal limits inferred from specific TARGs with the lowest FLP altitudes; 'This study (Stat)' denotes the altitudinal limits inferred from the statistics-based calculations of FLP altitudes. The limits from 'Field surveys' are based on the work of Qiu et al. (1983). The rows 'Difference (Min)' and 'Difference (Stat)' list the differences between the results inferred from this study and the field surveys, respectively. All the altitudes given by this study are derived from the SRTM digital elevation model data with an uncertainty of about 16 meters.

| | West of 86 °E | | East of 86 °E | |
| --- | --- | --- | --- | --- |
| | North-facing | South-facing | North-facing | South-facing |
| This study (Min) | 2490 m | 2583 m | 2895 m | 3088 m |
| This study (Stat) | 2775 m | 2831 m | 2998 m | 3038 m |
| Field surveys | 2730 m | 3050 m | 2900 m | 3250 m |
| Difference (Min) | -240 m | -467 m | -5 m | -162 m |
| Difference (Stat) | 45 m | -219 m | 98 m | -212 m |





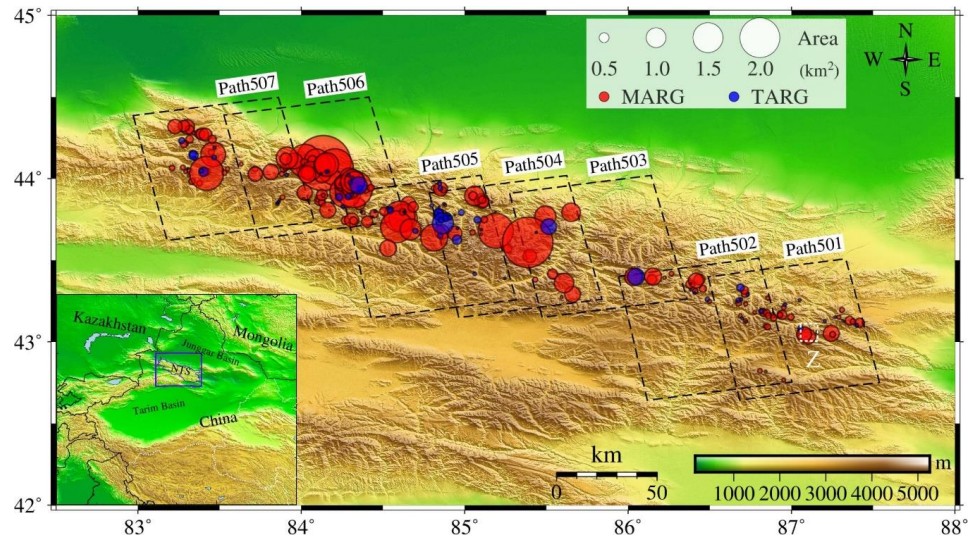

Figure 1. Topography map of the Northern Tien Shan (NTS) region. The black dashed boxes outline the footprints of seven frames of PALSAR frames along Path 501–507. The red and blue circles represent moraine-derived active rock glaciers (MARGs) and talus-derived active rock glaciers (TARGs), respectively. The circle size represents rock glacier area. The white dashed box (labeled as 'Z') is shown in Figure 3.





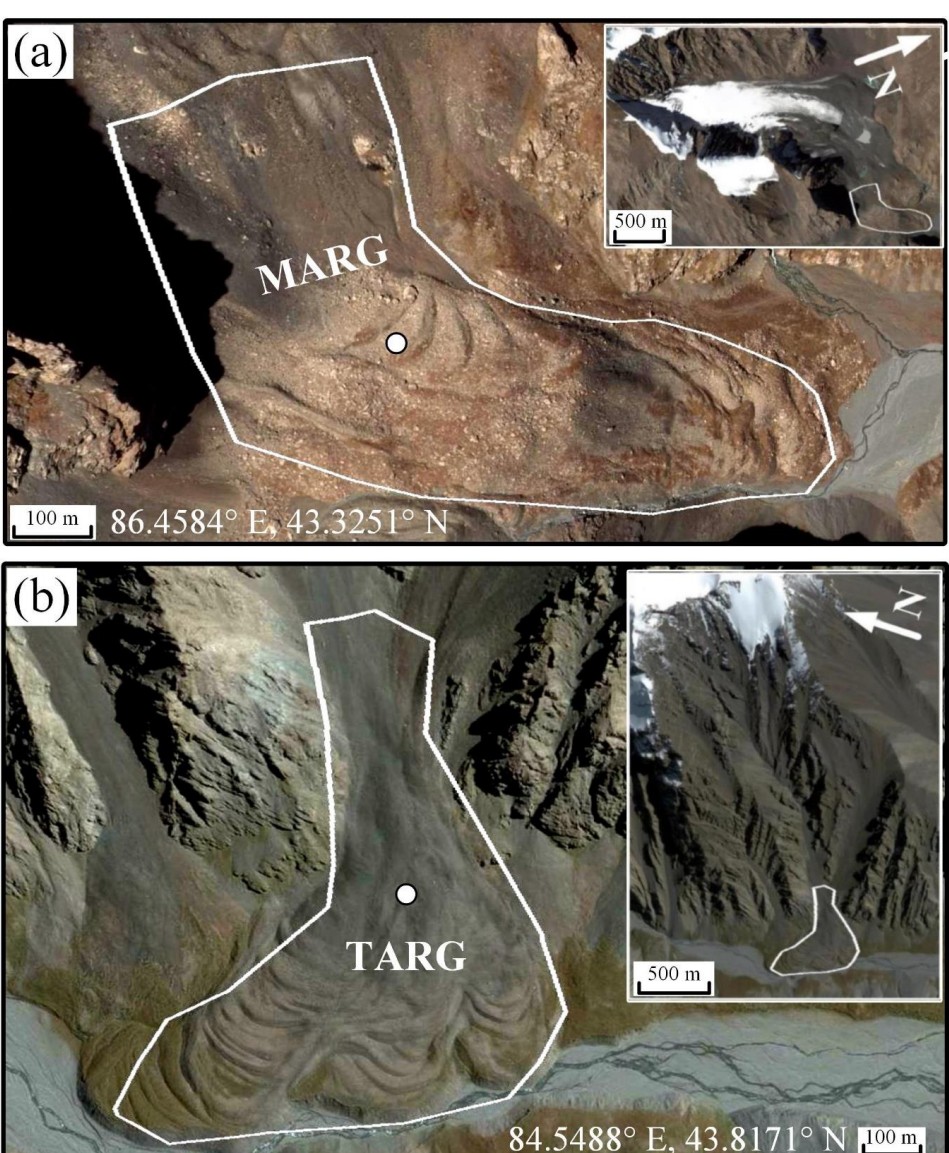

Figure 2. Google Earth images of (a) a typical moraine-derived active rock glacier (MARG) and (b) a talus-derived active rock glacier (TARG). The white dots represent the central location of the ARGs with their longitude and latitude given in the bottom of the figures. The inset images show the topographic and morphologic features surrounding the same rock glaciers.





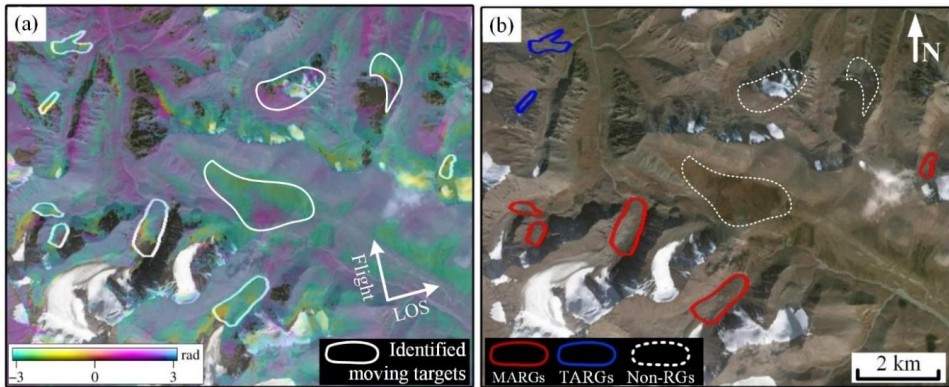

Figure 3. Example of the identified active rock glaciers in the zones 'Z' shown in Fig. 1. (a) shows the wrapped interferometric phase (in radians) for the pair 20070902-20071018 (Path 501). The solid white polygons denote the identified moving targets. (b) shows the corresponding Google Earth image, in which the red and blue polygons denote the identified MARGs and TARGs, respectively. The dashed white polygons denote moving targets that are not rock glaciers (labeled as 'Non-RGs').





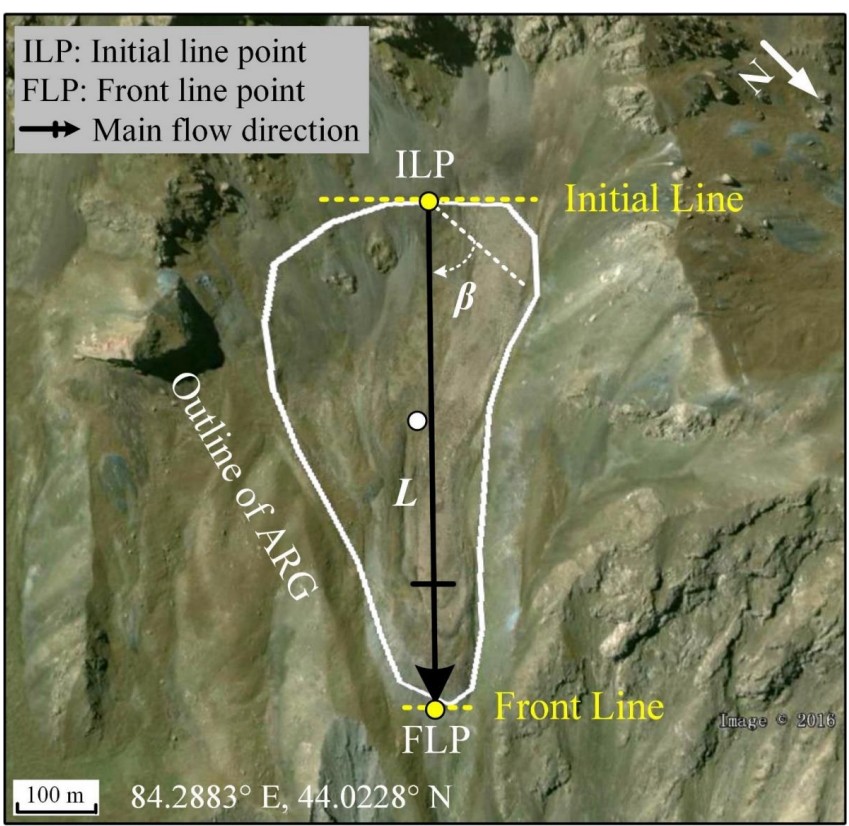

Figure 4. Google Earth image of a rock glacier, with labels showing its initial line point (ILP), front line point (FLP), length ($L$), and aspect angle ($\beta$). The longitude and latitude give the location of the central point (white dot) of the ARG.





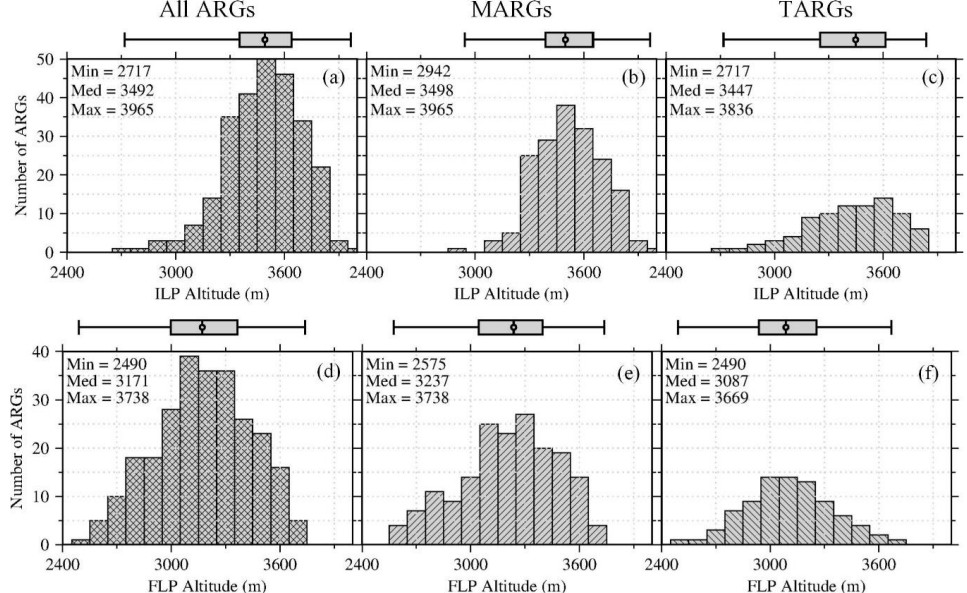

Figure 5. Histograms of the ILP altitudes (a–c) and FLP altitudes (d–f) for all ARGs, MARGs, and TARGs, respectively. The minimum (Min), median (Med), and maximum (Max) values for the ILP altitudes and FLP altitudes are denoted. Their units are meters. The box on the top of each sub-figure denotes the box-and-whisker symbol which gives the locations of minimum, the 25 % quantile, the 50 % quantile, the 75 % quantile, and the maximum values. The altitudes are derived from the SRTM digital elevation model data with an uncertainty of about 16 meters.





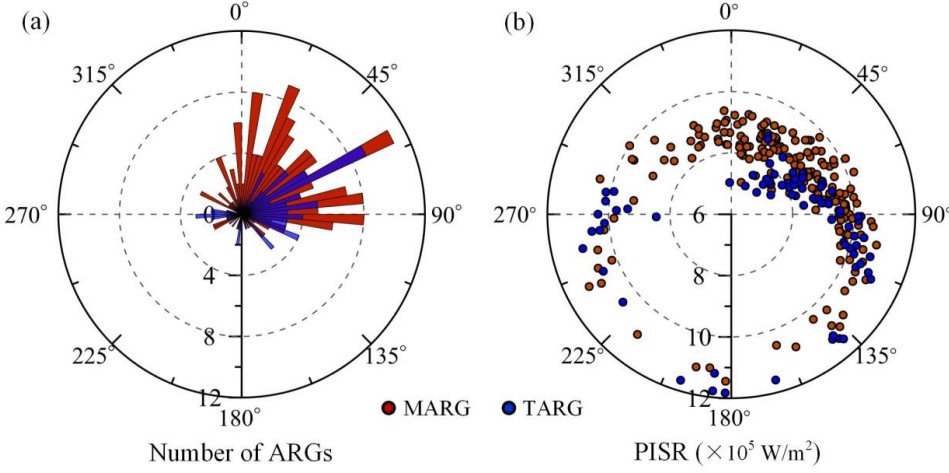

Figure 6. (a) Histogram of the aspect angles for the MARGs (red) and TARGs (blue), respectively.
(b) Wind rose plot shows the variations of potential income of solar radiation (PISR) with aspect
angles.





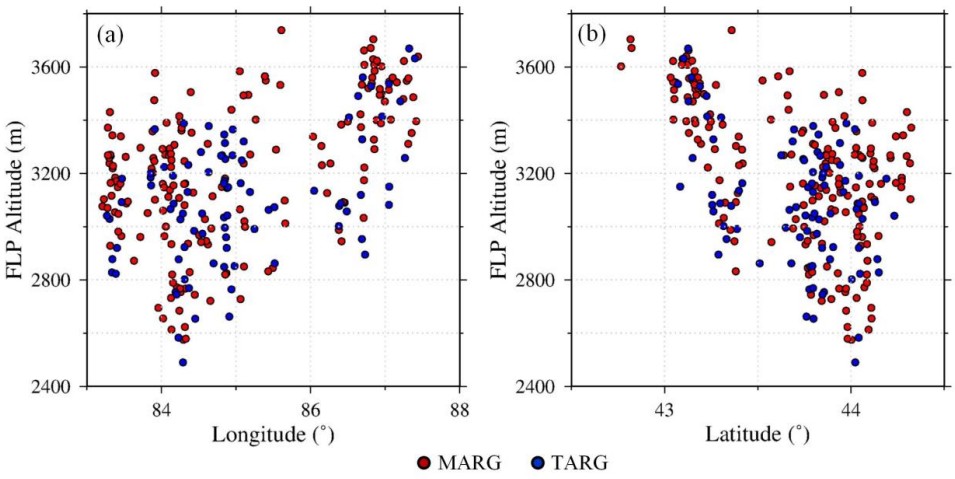

Figure 7. Scatter plots between the FLP altitudes and (a) the longitude, (b) latitude.





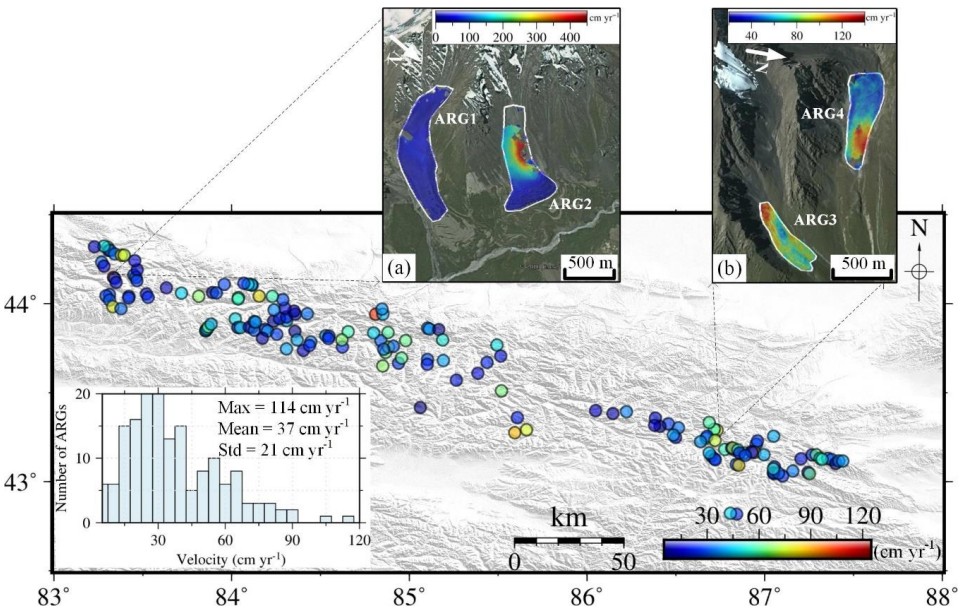

Figure 8. The bottom map shows the down-slope velocities of 170 ARGs measured from SAR interferometry, with the histogram shown in the inset. The top two figures show two pairs of ARGs adjacent to each other, with (a) and (b) showing the down-slope velocities. Note that different velocity color bars are used.





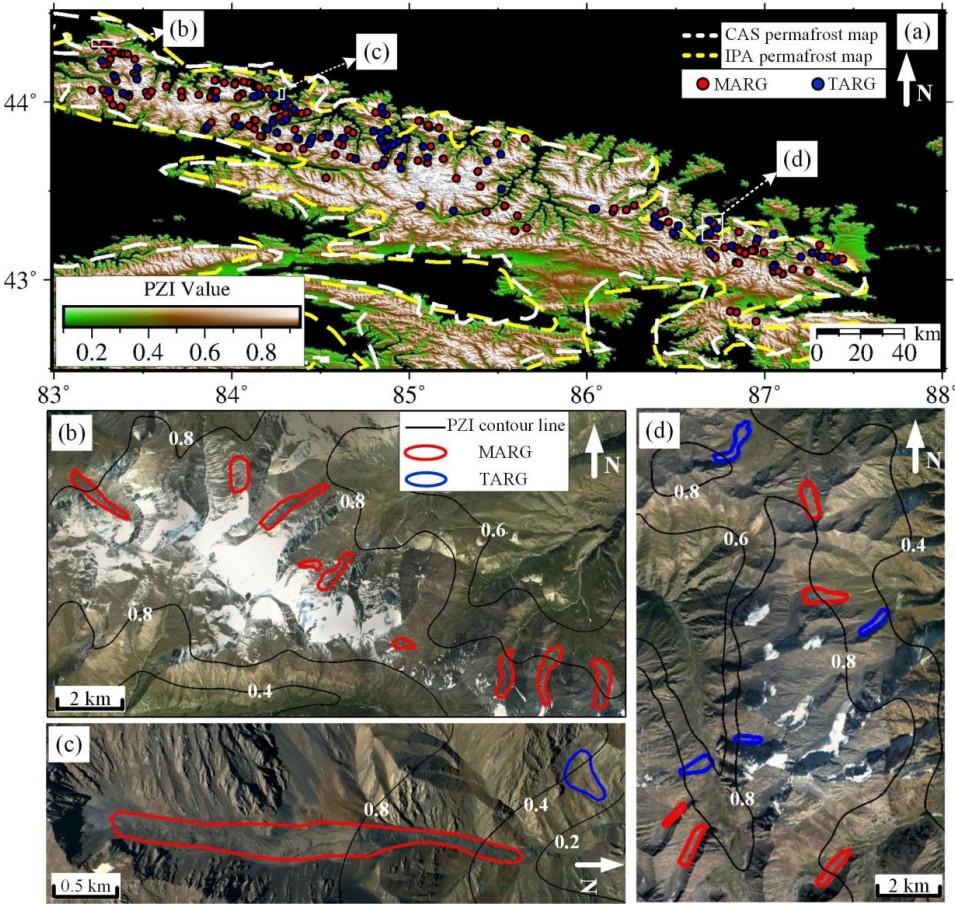

Figure 9. Comparison between the mapped ARGs and existing permafrost maps. The white and yellow dashed lines in (a) shows the mountain permafrost boundary of CAS permafrost map and discontinuous permafrost boundaries of IPA map, respectively. The green-to-white colors in (a) show the global Permafrost Zonation Index (PZI) values with areas with no PZI values are shown in black. The red and blue dots mark the locations of MARGs and TARGs, respectively. (b–d) show the local PZI maps for the three selected regions. (b) and (d) are selected randomly. (c) shows the MARG and TARG with the lowest FLP altitude in the entire study area. The thin black thin lines in (b-d) are contour lines of PZI values. The red and blue polygons outline the MARGs and TARGs, respectively.





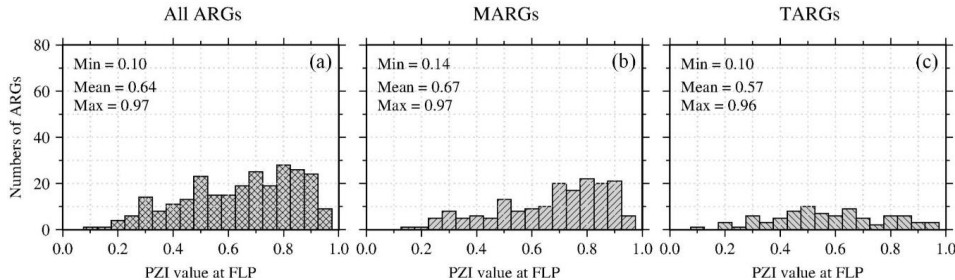

Figure 10. The histograms of the PZI values at the front line point (FLP) for all ARGs (a), MARGs (b), and TARGs (c), respectively.