# Peer review of "Mapping and inventorying active rock glaciers in the Northern Tien Shan of China using satellite SAR interferometry"

_The Cryosphere, 2016_

## Referee Comment (RC1) · G. Liu (Referee) · 5 Dec 2016

1. general comments - an initial paragraph or section evaluating the overall quality of the discussion paper After more than two decades silence, this manuscript offers us new knowledge, evidences and research methods about rock glaciers and permafrost distribution in the Northern Tien Shan of China. The paper based on 261 active rock glaciers which were recognized by combines SAR interferometry and optical images from Google Earth, give a detail discussing of their locations, geomorphic parameters, and down-slope velocities, and marking permafrost lower limit. This research is reference significance for alpine Periglacial landform research and permafrost mapping in the remote regions.

2. specific comments - addressing individual scientific questions/issues (1) recognition of rock glaciers: According to the manuscript and the supplement, most of the 261 active rock glaciers were correctly identified, yet some misreading appeared, for example, ARG 70, ARG 131 should be MARG. By the limitation of satellite data and research technique, a large number of small rock glaciers are not identified and compiled, especially the talus derived. For example, rock glaciers in the head water of Urumqi River reported by Cui and Zhu (1989), Zhu (1992), Zhu et al (1992), Liu et al (1995), not mentioned by the authors. ARG 94 was recognized as TARG, in the same way, site a (43.6429°N, 85.4292°E;) and b (43.6380°N, 85.4284°E) should be TARG. This greatly reduces the accuracy of number, regional and altitude distribution of the rock glaciers. This therefore, affect use of active rock glacier to determine permafrost lower limit. (2) identification of the initial line point (ILP) and front line point (FLP): It is not very clear how to determine ILP for talus derived rock glacier, for example ARG 22, 51; and moraine derived rock glacier, for example ARG 50, view Google Earth, the ILP located at moraine covered glacier. Some MARG mentioned in the manuscript, both ILP and FLP might be misreading, for example, ARG 95, the ILP is at glacier, the FLP should be down slope at 43.6226°N, 85.4043°E. ARG 219, 220, and 221 seem merge ARG 157, form a combined RG, the FLP might reach forest zone – see Google Earth. (3) surface velocity: It is better offering surface velocity by several years data. In the discussion, comparing with Cui and Zhu, Zhu. (4) indication of lower limit of permafrost: Though the manuscript give detailed discussion, the estimated lower limit of permafrost is well below field survey (Qiu et al, 1983) might be caused by missing the small RGs, especially the talus derived ones. (5) references: Page 12 4.4 references suggested: Jin HJ et al, 1993, Journal of Glaciology and Geocryology, 15(1). Qiu GQ, 1993, Journal of Glaciology and Geocryology, 15(1). Zhao L et al. 2010, Journal of Glaciology and Geocryology, 25(2).

3. technical corrections - a compact listing of purely technical corrections. technical corrections: typing errors, typographical corrections, etc. * page 3 line 77-78: "There is a lack of studies on surface velocities of rock glaciers or an inventory containing the

locations of the surveyed rock glaciers." See Cui and Zhu 1989. * page 4 line 97 (Zhu et al., 1992b) should add now reference. Zhao L et al. 2010. . . . . . * page 4 line 102 "(Cui and Zhu, 1989; Zhu et al., 1992a, 1992b; Liu et al., 1995)" - (Cui and Zhu, 1989; Zhu, 1992a, Zhu et al, 1992b; Liu et al., 1995) * page 6 line 151-152 "Debris-covered glaciers are usually covered with uniformly thin debris layer, whereas rock glaciers' debris cover is less homogenous and coarser." – what is the basis? * page 10 line 239 "IPLAs" ?! not mentioned in the text. * page 11-12, 4.3 Surface velocities of the active rock glaciers Reference Cui and Zhu, 1989 * page 17 line 442-444 The paragraph (4) seems could be omitted. * page 17 line 446-448 "This inventory offers a baseline dataset for the further investigations on permafrost modeling, slope stability, and water resource, etc." – why slope stability, and water resource? Not mentioned in the text.
* * *

---

## Referee Comment (RC2) · T. Strozzi (Referee) · 13 Dec 2016

The manuscript by Wang et al. nicely describes the application of SAR interferometry and Google Earth optical images to the mapping of active rock glaciers in the Northern Tien Shan. The structure of the paper is solid, the objectives of the work are clearly stated, the employed technology is well explained, the results are nicely described and illustrated, and the conclusions are well formulated. Although not completely novel, the approach of mapping and inventorying active rock glaciers from SAR interferometry and optical images is here throughout applied for the first time in the high mountains of western China, revealing new insights into rock glaciers as proxies of permafrost. The same approach presented by Wang et al. can be used for the systematic investiga-

tions of rock glaciers, and thus permafrost, over other remote mountainous locations. Considering the excellent work done by the authors in the redaction of the manuscript, I only have a few specific comments.

1. The short description about error sources in SAR interferograms at line 138 could be expanded a little bit to avoid the impression that their effects are not well considered or underestimated. In particular, the use of a coarse resolution DEM as SRTM results in uncompensated phase signals that scale up with the baseline of the interferograms. The authors state that maximum baselines considered in their studies are 600 m, but I would expect here a quantitative estimate of the phase disturbances that may occur in high mountains as a consequence of typical SRTM height errors or artefacts. In addition, a statement about the different distortions that occur with respect to the slope and orientation of the rock glaciers and the satellite line-of-sight direction should be included. Then, on the same paragraph, a short comment on the typical atmospheric disturbances at L-band with respect to the size of the active rock glaciers would be beneficial to strengthen the potential of the technology. Finally, Barboux et al. (2015) found out that phase unwrapping is the major limiting factor to the use of SAR interferometry for monitoring active rock glaciers in the Swiss Alps. In their paper, Wang et al. are not discussing at all possible phase unwrapping errors. I did some rapid calculations with the active rock glaciers velocities reported by the authors in the Northern Tien Shan. If maximum down-slope velocities of active rock glaciers in this region are about 114 cm yr-1, then the maximum line-of-sight velocities should be about 2/3 of the down-slope direction, i.e. about 76 cm yr-1. Over a time period of 46 days this would correspond to about 10 cm or less than one fringe a L-band. Indeed, in the Northern Tien Shan phase unwrapping is not a relevant issue, but this is not the case in many other mountain regions. Therefore, a short comment on phase unwrapping with respect to the velocities of the active rock glaciers should be included in the manuscript.

2. At line 63, the paper by Strozzi et al. (2010) is about landslides, not rock glaciers. Use instead Strozzi et al. (2004): Strozzi T., A. Kääb and R. Frauenfelder, Detecting

and quantifying mountain permafrost creep from in situ inventory, space-borne radar interferometry and airborne digital photogrammetry, Int. J. Remote Sensing, Vol. 25, No. 15, pp. 2919-2931, doi: 10.1080/0143116042000192330 2004.

3. Figure 9a is nearly impossible to interpret, there is too much information and there are too many colours and symbols. An alternative representation of this image should be proposed by the authors.
* * *

---

## Referee Comment (RC3) · T. Bolch (Referee) · 17 Jan 2017

Remarks on the manuscript by Wang et al. entitled

"Mapping and inventorying active rock glaciers in the Northern Tien Shan (China) using satellite SAR interferometry"

submitted to The Cryosphere

*General:*

The authors present a new rock glacier inventory for the Northeastern Tien Shan which they derived from InSAR data and Google earth imagery. Moreover they analyze the topographic characteristics of the rock glaciers and use the rock glacier inventory to estimate the permafrost distribution in the study region.

The approach to map the creeping permafrost features is interesting and promising and has rarely been applied so far. The inventory is the first of its kind in Northwestern China and the topographic and climatic analysis provide useful insights. I (and surely the scientific community also) appreciate very much that the inventory is made available as supplementary material.

However, there are some issues which need to be improved from my point of view before the manuscript can be published.

The most important issues are:

- The delineation of the rock glaciers seems to be unprecise as e.g. visible from the figures: The upper boundary may be hard to define precisely but some more efforts should be made; is it not clear from Fig. 2 which criteria was used and the boundary is very probably not rectangular. It seems to me from Figs. 2b and 4 that exposed rocks are included. The lower boundary visible in Fig. 2b is also too rough as parts which clearly belong to the rock glacier are missed while others (e.g. even a part of the river were included). This needs to be improved as good as possible for all outlines so that at least no obvious errors are made.

- Rock glaciers provide a hint where the lower limit of permafrost is located. However, the lower limit of rock glaciers varies significant due to topographic factors and the blocky material favours cooler temperatures allowing rock glaciers to exist at elevations where permafrost is otherwise unlikely (see. e.g. modelling studies for the Northern Tien Shan in Kazakhstan/Kyrgyzstan, see references). I'd therefore suggest rather to discuss the suitability of rock glaciers for investigation the permafrost occurrence (as you partly already did) than presenting new results of a lower limit of the permafrost occurrence which is highly uncertain. The authors should also keep in mind that there is no clear limit but that especially in mountains the permafrost occurrence is very heterogeneous.

- There is limited discussion where the authors put their approach and results into context. There are several studies existing which were presenting rock glacier inventories and further information in the neighbouring Kazakh and Kyrgyz Tien Shan. Several ones are in Russian which might be difficult to understand (but may still be considered as they provide valuable information) but there are also several published in English (especially by A. Gorbunov, see references). One more recent paper co-authored by Gorbunov presents also a topographic analysis which would be very interesting to compare your results to.

- Recent findings show a clear seasonal behaviour of the surface velocity (e.g. Wirz et al. 2016). Hence, it is important to mention the acquisition period of the data used to calculate the velocity also in the text. The seasonal effect should also be considered in the discussion.

- With respect to the discussion I suggest to include a separate discussion section where you put your approach and results into context of the existing literature. Currently, the discussion section contains only a comparison of the rock glacier derived permafrost estimates to permafrost distribution estimates.
- A minor but important point: The Tien Shan is a large mountain range stretching from Uzbekistan into China. Considering the Tien Shan as a whole using "Northern Tien Shan" is not correct. You should rather use "Northeastern Tien Shan" in the title and elsewhere.

Specific comments:

*Abstract:*

L10: Rock glaciers are widespread not only in western China but in the whole of Tien Shan.

L11: There are few recent studies, but your statement is true for Western China. Please correct.

L24: The approach is interesting for global rock glacier mapping and not only for western China.

**1. *Introduction:**

L28: I am missing the classical monography by Barsch (Barsch 1996) in the references. I'd rather cite Haeberli et al. (2006) instead of 2010 here.

L36: Not sure if a definition of inactive and relict rock glaciers are needed here. This is a cryospheric journal where the readers should know such basic knowledge.

L45: Brenning (2005) already mentioned it and used the term intact for active and inactive rock glaciers. You would are able to distinguish which is a major advantage.

L47ff: You should mention also somewhere that multi-temporal optical images can and were also used to investigate rock glacier velocity (e.g. Gorbunov et al. 1992, by visual interpretation; Kääb et al. 1997 by feature tracking).

L68: Mention here at least one of the existing studies.

L74: Include here few of the existing studies of the Kazakh and Kyrgyz Tien Shan.

L80: You may mention here the study by Bolch and Gorbunov for the nearby Northern Tien Shan in Kazakhstan and Kyrgyzstan and maybe also Schmid et al. (2015)

**2. *Study area:**

General: Provide more details about the general characteristics of the Tien Shan, the subdivision, and especially the climatic conditions.

L97: Be more specific about the annual precipitation amount. The west is wetter than the east but not really wet. The climate is only relative humid for the continental conditions.

L102: You cite here almost the same studies than in L75 in the introduction. I suggest to cite in the intro those which provide information about the larger regions or from surrounding ranges and in the study region section the specific ones.

**3. *Methodology:**

General: The other reviewers are more experts in SAR processing. Therefore I will not comment on the technical aspects here. However, I'd like to see a better figure where the authors present the identification of moving surface based on the wrapped interferometric phase. Fig. 3 is interesting in this regard but I find it hard to understand how you identified surface displacements based on this image. Maybe a larger image or a zoom would help.

Provide a short information about the quality and suitability of the images available at the time of the study in google earth, e.g. had all images good snow conditions and where all of very high resolution?

L148: I agree that active rock glaciers have usually little or no vegetation, but not always (e.g. there are trees on rock glaciers in other parts of the Tien Shan). Hence, write "have usually little or no vegetation" or similar.

L151: It is not true that debris-covered glaciers "are usually covered with uniformly thin debris layer". There are manifold studies which show that the debris thickness usually increases towards the terminus and that the surface of a debris-covered glaciers is characterised by ice cliffs and supraglacial lakes. It is partly hard to distinguish clearly between debris-covered glaciers and rock glaciers as gradual transitions to moraine-derived rock glaciers exist especially in continental conditions.

L161: What is a rooting zone "Z" and where is it in Fig. 1?

L184f: I find the abbreviations ILP (initial line point) and FLP (front line point) not intuitive as you are mainly interested in the altitude. Humlum (1998) uses RILA (rock glacier initiation line altitude). As you are referring to the maximum and minimum altitude I'd use $h_{mim}$ and $h_{max}$. But you may decide.

L197f: It can be quite difficult to identify the upper boundary. Hence, InSAR is quite promising. Provide more details and examples for how you identified the upper boundary. Ridges and furrows are typical for compressive flow in the lower rock glacier area and, hence, you might have missed parts if you only use these criteria.

L200: Should be Gruber (2012).

L206: Why do you mention "lower limit of the permafrost distribution" here. You are quantifying the rock glacier parameters.

*4. Results:*

L222f: Move to methods.

L250ff: What is about the influence of the general topography? It could be worth to compare the aspects of the rock glaciers to the aspect distribution of the investigated mountain ranges. The aspect distribution of the rock glaciers in Kazakh and Kyrgyz Northern Tien Shan is clearly influences by the topography.

L252: Kaldybayev et al. (2016) investigate glaciers in Dzhungar Alatau which is close to you study region but is not the Northern Tien Shan.

L266: Be careful with the statement about the lower limit. See my general comment above.

L270: What is about the precipitation? I would assume the precip is also of importance.

L278f: This is an important point and should be discussed more in detail.

L280ff: You need to consider that the temperature in the blocky material of the rock glaciers can be significant colder than the MAAT (e.g. Gorbunov et al. 2004).

L287ff: The first lines of this section describe methods and should be presented in the methods section. In addition, as mentioned information about the time of the year when the velocity was measured are required.

L302f: The presence of water (e.g. from snow melt or heavy rain fall) has a strong influence on the short term variation. This should be mentioned and discussed along with the acquisition period of the data.

*5. Discussion*

I suggest a separate discussion section where you put your results into context. I suggest to slightly shorten the discussion about the comparison to the existing permafrost maps.

*6. Conclusions*

Readers often read the abstract and look at the figures conclusion only before they decide to read the entire paper. I would therefore not use non-common abbreviations in the conclusions and figure captions.

L434: Use one decimal digit only. The exact area is quite uncertain.

L442ff: Conclusion 4 is hard to understand.

L446: The methods cannot only applied in China…

L450: Reformulate the last sentence. Rock glaciers provide a hint for permafrost occurrence but should be used with care when modelling permafrost distribution.

Do not hesitate to ask in case you have any questions to my review or need additional information.

Best regards,

Tobias Bolch

*References:*

Barsch, D. (1996). Rock Glaciers. Indicators for the Present and Former Geoecology in High Mountain Environments. Berlin: Springer.

Bolch, T., & Gorbunov, A.P. (2014). Characteristics and Origin of Rock Glaciers in Northern Tien Shan (Kazakhstan/Kyrgyzstan). Permafrost and Periglacial Processes, 25, 320–332.

Brenning, A. (2005). Geomorphological, hydrological and climatic significance of rock glaciers in the Andes of Central Chile (33-35°S). Permafrost and Periglacial Processes, 16, 231–240.

Gorbunov, A.P. (1983). Rock glaciers in the mountains of Middle Asia. In: 4th International Conference on Permafrost (pp. 359–362): National Academy Press, Washington, DC.

Gorbunov, A.P., Titkov, S.N., & Polyakov, V. (1992). Dynamics of the Rock Glaciers of the Northern Tien Shan and the Djungar Alatau, Kazakhstan. Permafrost and Periglacial Processes, 3, 29–39.

Gorbunov, A.P., Marchenko, S., & Severskiy, E.V. (2004). The thermal environment of blocky materials in the mountains of Central Asia. Permafrost and Periglacial Processes, 15, 95–98.

Kääb, A., Haeberli, W., & Gudmundsson, G. (1997). Analysing the creep of mountain permafrost using high precision aerial photogrammetry: 25 years of monitoring Gruben rock glacier, Swiss Alps. Permafrost and Periglacial Processes, 8, 409–426.

Marchenko, S.S. (2001). A model of permafrost formation and occurrences in the intracontinental mountains. Norsk Geografisk Tidsskrift - Norwegian Journal of Geography, 55, 230–234.

Schmid, M.-O., Baral, P., Gruber, S., Shahi, S., Shrestha, T., Stumm, D., & Wester, P. (2015). Assessment of permafrost distribution maps in the Hindu Kush Himalayan region using rock glaciers mapped in Google Earth. The Cryosphere, 9. http://www.the-cryosphere.net/9/2089/2015/.

Titkov, S.N. (1988). Rock Glaciers and glaciation of the Central Asian Mountains. In Proceedings of the 5th International Permafrost Conference, Vol. 1 (pp. 259–262): Tapir Publishers.

Wirz, V., Gruber, S., Purves, R.S., Beutel, J., Gärtner-Roer, I., Gubler, S., & Vieli, A. (2016). Short-term velocity variations at three rock glaciers and their relationship with meteorological conditions. Earth Surf. Dynam., 4, 103–123.

---

## Author Comment (AC1) · 13 Feb 2017

We thank all the reviewers for their constructive comments. We have addressed them all. Please refer to the following linked file for our revised manuscript with changes highlighted in yellow.

Please also note the supplement to this comment:
http://www.the-cryosphere-discuss.net/tc-2016-254/tc-2016-254-AC1-supplement.pdf

---

## Author Comment (AC2) · 13 Feb 2017

**Reply to comments by G. Liu on "Mapping and inventorying active rock glaciers in the Northern Tien Shan (China) using satellite SAR interferometry"**

**General comments**

After more than two decades silence, this manuscript offers us new knowledge, evidences and research methods about rock glaciers and permafrost distribution in the Northern Tien Shan of China. The paper based on 261 active rock glaciers which were recognized by combines SAR interferometry and optical images from Google Earth, give a detail discussing of their locations, geomorphic parameters, and down-slope velocities, and marking permafrost lower limit. This research is reference significance for alpine Periglacial landform research and permafrost mapping in the remote regions.

We thank Prof. G. Liu for his positive comments on our approach and results. We also appreciate his careful consideration and detailed comments. Our replies are highlighted in blue.

**Specific comments**

(1) Recognition of rock glaciers:

- According to the manuscript and the supplement, most of the 261 active rock glaciers were correctly identified, yet some misreading appeared, for example, ARG 70, ARG 131 should be MARG.

Authors: Thanks for your correction. We have accordingly corrected the types of ARG70 and ARG131 as MARGs. We also revised the relevant statistics related to the MARGs and TARGs through the paper.

- By the limitation of satellite data and research technique, a large number of small rock glaciers are not identified and compiled, especially the talus derived. For example, rock glaciers in the head water of Urumqi River reported by Cui and Zhu (1989), Zhu (1992), Zhu et al (1992), Liu et al (1995), not mentioned by the authors.

Authors: It is true that we may have missed some small rock glaciers because the InSAR phase maps have a moderate spatial resolution of 15 m. We have highlighted this potential limitation of our method in Section 4 (see Lines 259–268).

Following the reviewer's suggestion, we have added a discussion subsection to compare our inventory with the previous studies on rock glaciers in the Tien Shan (Cui and Zhu, 1989; Gorbunov et al. 1992; Zhu 1992; Zhu et al.1992; Liu et al. 1995). We have reviewed and compared the altitude distribution, geomorphic descriptions and surface velocities of the

rock glaciers that have been documented in these studies with our results (see Lines 409–435). However, we can only perform a qualitative, not quantitative, comparison. This is because (1) the exact locations of the rock glaciers documented in these studies are not available; (2) the method we adopted to map active rock glaciers is significantly different from the method used in these previous studies, making the comparison impractical.

- ARG 94 was recognized as TARG, in the same way, site a (43.6429◦N, 85.4292◦E ;) and b (43.6380◦N, 85.4284◦E) should be TARG. This greatly reduces the accuracy of number, regional and altitude distribution of the rock glaciers. This therefore, affect use of active rock glacier to determine permafrost lower limit.

Authors: We have investigated the two locations of site 'a' (43.6429˚N, 85.4292˚E) and 'b' (43.6380˚N, 85.4284˚E) on the Google Earth images. From the geomorphic features, we interpret that these two sites host two talus-derived rock glaciers. However, we found that our InSAR phase map suffers from severe de-correlation problem at these two locations, likely due to radar shadows. Therefore, we excluded them from our inventory. We have pointed out that the number of ARGs we compiled is a conservative estimation of all the ARGs in the NTS, and listed several reasons for this kind of conservative estimation (please see Lines 259–268). This is also why we gave a range of estimations rather than exact values of the lower altitude limits of permafrost, as described and justified in Section 4.4. Nevertheless, as discussed in Section 5.3, the distribution of the ARGs we inventoried are in good agreement with these three existing permafrost maps in the Northern Tien Shan.

(2) Identification of the initial line point (ILP) and front line point (FLP):

- It is not very clear how to determine ILP for talus derived rock glacier, for example ARG22, 51; and moraine derived rock glacier, for example ARG 50, view Google Earth, the ILP located at moraine covered glacier.

Authors: As described in the first paragraph in Section 3.3 (Lines 220–230), we determined the initial line point (ILP) and front line point (FLP) of a rock glacier based on both the InSAR phase measurements and the geomorphological features from the Google Earth images. The variations of the InSAR phase represent ground movements, and thus we can outline an active rock glacier (ARG) based on the InSAR phase variations. We interpreted the active boundary in the rooting zone of an ARG as the initial line where the permafrost starts to creep. We then used the central point of the initial line as the ILP. The FLP of the ARG can be determined based on the similar method. In addition, the FLP is generally located in the lowest place where a rock glacier can reach, which can be easily identified from the Google Earth images. For example, Fig. R1(a) below shows the outline of the ARG22 determined from the InSAR phase variations. Fig. R1(b) shows the corresponding

Google Earth image. We have rewritten the relevant text in Section 3.3 to clarify our method for identifying the ILP and FLP (see Lines 220–230).

[Figure]

**Fig. R1** ARG22 and its ILP and FLP. (a) shows the InSAR phase map for the ARG22, and the blue line denotes the active boundary determined from the phase variations. (b) shows the Google image.

- Some MARG mentioned in the manuscript, both ILP and FLP might be misreading, for example, ARG 95, the ILP is at glacier, the FLP should be down slope at $43.6226^{\circ}$ N, $85.4043^{\circ}$ E.

Authors: Thank you very much for your corrections. We have corrected the ILP and FLP for the ARG 95 based on the interpretation of historical images from the Google Earth. We have also checked the ILP/FLP for each identified ARG to improve the accuracy of our inventory.

- ARG 219, 220, and 221 seem merge ARG 157, form a combined RG, the FLP might reach forest zone – see Google Earth.

Authors: Some nearby ARGs could merge and form a combined rock glacier. In such cases, we classified each sub-ARG as individual rock glacier for following two reasons. (1) The sub-ARGs have different aspect angles. (2) We want to include the identified ARGs as many as possible. We have added one sentence to explain the combined ARGs in our revised manuscript, please see Lines 187-189.

(3) Surface velocity: It is better offering surface velocity by several years' data. In the discussion, comparing with Cui and Zhu, Zhu.

Authors: It is a good idea to map the surface velocities of ARGs using several year's SAR

data. The multi-temporal observations would help us to reveal the seasonal fluctuations of the velocities of the ARGs (e.g., Liu et al., 2013). However, due to the rough topography in the NTS area, very few ALOS PALSAR interferometric pairs are suitable for InSAR time series analysis. The suitable interferometric pairs should have both small temporal span and perpendicular baseline, thus maintaining good phase coherence. In this study, we selected the PALSAR interferograms that have a temporal span of 46/92 days and perpendicular baselines smaller than 600 m. In addition, we only used the PALSAR images acquired in summer to form the interferograms except for the Path 503. We have added some descriptions about the SAR data we used in this study, please see Lines 136–140.

We have added a sub-section (Section 5.1) in the discussion part to compare our inventory with the previous studies. In which, we compared the surface velocities derived from our InSAR measurements with the field observations conducted by Cui and Zhu (1989) and Zhu (1992a). Please see Lines 409–435.

(4) Indication of lower limit of permafrost: Though the manuscript give detailed discussion, the estimated lower limit of permafrost is well below field survey (Qiu et al., 1983) might be caused by missing the small RGs, especially the talus derived ones.

Authors: Thank you for pointing out this possible reason that would influence the lower limit of permafrost we estimated. The InSAR phase maps we generated have a spatial resolution of about 15 meters. It is possible that some small rock glacier in the shadows of radar images cannot be identified. We have added this possible reason in the revised version (see Lines 259–268).

(5) References: Page 12, 4.4, references suggested: Jin HJ et al, 1993, Journal of Glaciology and Geocryology, 15(1). Qiu GQ, 1993, Journal of Glaciology and Geocryology, 15(1). Zhao L et al. 2010, Journal of Glaciology and Geocryology, 32(2).

Authors: Thanks for your suggestions. We have added these three references in the revised manuscript, please see Lines 372, 405–407.

**3. Technical corrections: a compact listing of purely technical corrections, typing errors, typographical corrections, etc.**

(1) Page 3 line 77-78: "There is a lack of studies on surface velocities of rock glaciers or an inventory containing the locations of the surveyed rock glaciers." See Cui and Zhu 1989.

Authors: The relevant sentence has been rewritten as: "The studies on surface velocities of rock glaciers are very limited except for the field surveys at several specific sites conducted by Cui and Zhu (1989) and Zhu (1992a)."

(2) Page 4 line 97 (Zhu et al., 1992b) should add now reference. Zhao L. et al. 2010.

Authors: The reference of Zhao et al., (2010) has been added, please see Line 110.

(3) Page 4 line 102 "(Cui and Zhu, 1989; Zhu et al., 1992a, 1992b; Liu et al., 1995)" - (Cui and Zhu, 1989; Zhu, 1992a, Zhu et al, 1992b; Liu et al., 1995)

Authors: The citation format has been corrected in the revised version.

(4) Page 6 line 151-152 "Debris-covered glaciers are usually covered with uniformly thin debris layer, whereas rock glaciers debris cover is less homogenous and coarser." – What is the basis?

Authors: Considering the comments of the referee 3 (Dr. Bolch), we have rewritten the relevant sentences as follows (please see Lines 174–180):

"We distinguished ARGs from debris-covered glaciers based on the de-correlation conditions of interferograms and their different visual features on the Google Earth images. Compared with rock glaciers, debris-covered glaciers generally move much faster (Janke et al., 2015), which results in large areas of de-correlation in our PALSAR interferograms. The surface of a debris-covered glacier is usually characterized by ice cliffs and supra-glacial lakes (Bolch et al., 2007). And the rooting zones of debris-covered glaciers are continuous with clean glacier ice (Davies et al., 2013; Lukas et al., 2007)."

(5) Page 10 line 239 "IPLAs"? Not mentioned in the text.

Authors: This is a typo. "IPLAs" has been changed to "ILPAs".

(6) Page 11-12, 4.3 Surface velocities of the active rock glaciers Reference Cui and Zhu, 1989.

Authors: We have added a discussion subsection (Section 5.1) to compare our inventory with the previous studies of rock glaciers in the Tien Shan (Lines 409–435).

(7) Page 17 line 442-444. The paragraph (4) seems could be omitted.

Authors: We have removed the paragraph (4).

(8) Page 17 line 446-448 "This inventory offers a baseline dataset for the further investigations on permafrost modeling, slope stability, and water resource, etc." – why slope stability, and water resource? Not mentioned in the text.

Authors: We have added some descriptions and references in the "Introduction" part to clarify the significance of rock glaciers on the water resource in the semi-arid region and the potential hazards in the high mountains. Rock glaciers serve as important freshwater reserves because of the potential melting of their internal ice in the runoff season. Active rock glaciers

may cause mass waste hazards on steep slopes when they move downslope, and thus threatening the sensible infrastructures in some downhill areas (Schoeneich et al., 2015). Please see Lines 31–33, 37–38.

---

## Author Comment (AC3) · 13 Feb 2017

**Reply to comments by T. Strozzi on "Mapping and inventorying active rock glaciers in the Northern Tien Shan (China) using satellite SAR interferometry"**

**1. General comments**

The manuscript by Wang et al. nicely describes the application of SAR interferometry and Google Earth optical images to the mapping of active rock glaciers in the Northern Tien Shan. The structure of the paper is solid, the objectives of the work are clearly stated, the employed technology is well explained, the results are nicely described and illustrated, and the conclusions are well formulated. Although not completely novel, the approach of mapping and inventorying active rock glaciers from SAR interferometry and optical images is here throughout applied for the first time in the high mountains of western China, revealing new insights into rock glaciers as proxies of permafrost. The same approach presented by Wang et al. can be used for the systematic investigations of rock glaciers, and thus permafrost, over other remote mountainous locations. Considering the excellent work done by the authors in the redaction of the manuscript, I only have a few specific comments.

We thank Dr. T. Strozzi for his thoughtful consideration and helpful comments on our manuscript. We have addressed all the comments below. Our replies are highlighted in blue.

**2. Specific comments**

1. The short description about error sources in SAR interferograms at line 138 could be expanded a little bit to avoid the impression that their effects are not well considered or underestimated. In particular, the use of a coarse resolution DEM as SRTM results in uncompensated phase signals that scale up with the baseline of the interferograms. The authors state that maximum baselines considered in their studies are 600 m, but I would expect here a quantitative estimate of the phase disturbances that may occur in high mountains as a consequence of typical SRTM height errors or artefacts.

Authors: We have added a sentence copied below to demonstrate the contribution of topographic error to the interferometric phase (please see Lines 151–156):

"The contribution of the topographic error to the interferometric phase is proportional to the perpendicular baseline, the radar slant range, and the radar look angle (Rosen et al., 2000). The maximum perpendicular baseline of the interferograms produced in this study is about 600 m. Adopting the vertical accuracy of the digital elevation model as its nominal value of 16 m (Farr et al., 2004), we estimate that the residual topographic phase in the InSAR measurements would be about 0.92 radians, corresponding to 1.7 cm.

In addition, a statement about the different distortions that occur with respect to the slope

and orientation of the rock glaciers and the satellite line-of-sight direction should be included.

Authors: We have rewritten the relevant part in Section 4 (copied below) to include the influences of geometric distortions to the identification of ARGs (Please see Lines 259–268):

"The number of ARGs we compiled is a conservative estimation of all the ARGs in the NTS due to the following three reasons. First, we may have missed some ARGs due to the geometric distortions such as shadows and layovers in the SAR images. For the PALSAR images used in this study, the rock glaciers facing east with the slope angles larger than 51.3 ° are in the radar shadows, and the rock glaciers facing west with slope angles larger than 38.7 °are in the layover regions. These distortions would result in significantly phase de-correlation and make it difficult to identify rock glaciers in these regions. Second, ARGs facing nearly north or south might also have been missed, as InSAR (using images from all space missions, including ALOS) is not sensitive to ground motions along these directions. Finally, some small ARGs could not be identified as the interferogram maps have a moderate resolution of about 15 m."

Then, on the same paragraph, a short comment on the typical atmospheric disturbances at L-band with respect to the size of the active rock glaciers would be beneficial to strengthen the potential of the technology.

Authors: We have added two sentences (copied below) to describe the atmosphere disturbance on the L-band SAR interferometry, with concerning the size of the active rock glaciers, please see Lines 156–159.

"The atmospheric (including tropospheric and ionospheric) effects are generally manifest themselves as long wavelength signals on the order of 1–10 km in interferogram maps (Hanssen, 2001), thus can be assumed nearly constant over a specific rock glacier. By using a local reference point just outside each rock glacier (see section 3.2), we effectively removed these large-scale atmospheric errors."

Finally, Barboux et al. (2015) found out that phase unwrapping is the major limiting factor to the use of SAR interferometry for monitoring active rock glaciers in the Swiss Alps. In their paper, Wang et al. are not discussing at all possible phase unwrapping errors. I did some rapid calculations with the active rock glaciers velocities reported by the authors in the Northern Tien Shan. If maximum down-slope velocities of active rock glaciers in this region are about 114 cm yr$^{-1}$, then the maximum line-of-sight velocities should be about 2/3 of the down-slope direction, i.e. about 76 cm yr$^{-1}$. Over a time period of 46 days this would correspond to about 10 cm or less than one fringe an L-band. Indeed, in the Northern Tien

Shan phase unwrapping is not a relevant issue, but this is not the case in many other mountain regions. Therefore, a short comment on phase unwrapping with respect to the velocities of the active rock glaciers should be included in the manuscript.

Authors: We have added some words to demonstrate the unwrapping issues for addressing the surface velocities of rock glaciers using InSAR (Please see Lines 205–210). We also carefully examined unwrapped interferograms and confirmed no unwrapping errors over the rock glaciers we mapped.

"Previous studies indicated that phase unwrapping is the major limiting factor to the use of SAR interferometry for monitoring active rock glaciers (Barboux et al., 2015). Phase unwrapping may fail in the areas where phase gradients are large due to fast slope movements. To minimize the phase unwrapping errors, we applied the phase filtering with a window of $8 \times 8$ pixels and masked out the decorrelated areas with a coherence threshold of 0.3 before phase unwrapping."

2. At line 63, the paper by Strozzi et al. (2010) is about landslides, not rock glaciers. Use instead Strozzi et al. (2004): Strozzi T., A. Kääb and R. Frauenfelder, Detecting and quantifying mountain permafrost creep from in situ inventory, space-borne radar interferometry and airborne digital photogrammetry, Int. J. Remote Sensing, Vol. 25, No. 15, pp. 2919-2931, doi: 10.1080/0143116042000192330 2004.

Authors: We have changed the reference "Strozzi et al. (2010)" to "Strozzi et al. (2004)" in Line 65.

3. Figure 9a is nearly impossible to interpret, there is too much information and there are too many colors and symbols. An alternative representation of this image should be proposed by the authors.

Authors: We have broken Figure 9a into two sub-figures to improve interpretation. We also changed Figures 9(c–d) as a new Figure 10.

---

## Author Comment (AC4) · 13 Feb 2017

**Reply to comments by Tobias Bolch on "Mapping and inventorying active rock glaciers in the Northern Tien Shan (China) using satellite SAR interferometry"**

We thank Dr. Bolch for his detailed and insightful review of the discussion paper. We have addressed all the comments and made the suggested changes in the revised version of our manuscript. Our point-by-point replies (in blue) to the critical comments (in black) are listed below.

The most important issues are:

- The delineation of the rock glaciers seems to be unprecise as e.g. visible from the figures: The upper boundary may be hard to define precisely but some more efforts should be made; is it not clear from Fig. 2 which criteria was used and the boundary is very probably not rectangular. It seems to me from Figs. 2b and 4 that exposed rocks are included. The lower boundary visible in Fig. 2b is also too rough as parts which clearly belong to the rock glacier are missed while others (e.g. even a part of the river were included). This needs to be improved as good as possible for all outlines so that at least no obvious errors are made.

Authors: We are grateful for the suggestions. We have checked the outline of each inventoried ARG and re-delineated some of them based on the InSAR phase and optical images. We also re-plotted Fig. 2 and Fig. 4 to correct the errors pointed out by the reviewer.

- Rock glaciers provide a hint where the lower limit of permafrost is located. However, the lower limit of rock glaciers varies significant due to topographic factors and the blocky material favors cooler temperatures allowing rock glaciers to exist at elevations where permafrost is otherwise unlikely (see. e.g. modelling studies for the Northern Tien Shan in Kazakhstan/Kyrgyzstan, see references). I'd therefore suggest rather to discuss the suitability of rock glaciers for investigation the permafrost occurrence (as you partly already did) than presenting new results of a lower limit of the permafrost occurrence which is highly uncertain. The authors should also keep in mind that there is no clear limit but that especially in mountains the permafrost occurrence is very heterogeneous.

Authors: Thanks very much for the insightful suggestion.

The purpose of this study is to propose a new method for inventorying active rock glaciers in the periglacial high mountains. We chose the Northern Tien Shan in China as a study area and then used the inventoried talus-derived active rock glaciers (TARGs) to infer the lower limit of permafrost. As suggested by the reviewer, there may be a high uncertainty in the estimates of permafrost limits due to the heterogeneous thermal region in mountainous environment. Previous studies conducted by Gorbunov et al. (2004) in Transili Alatau Tien Shan revealed a significant difference between the thermal regime of coarse blocky materials

and adjacent fine-grained soils. Mean annual temperatures inside the coarse debris could be 2.5–4.0 °C cooler than mean annual air temperatures (MAATs), which could cause rock glaciers occur at elevations where permafrost is unlikely to exist. However, the statistics of our inventory show that the MAAT at the TARGs is -5.4 °C, and 83 % of the TARGs have the MAAT lower than -4 °C. This indicates that the TARGs we used for estimating the lower limit of permafrost are in a cold environment, and thus reinforcing the credibility of the estimates. We now include such analysis and interpretation in the revised manuscript (Lines 456–459).

It is indeed important to discuss the suitability of rock glaciers for investigating the permafrost occurrence. An accurate permafrost distribution map will be helpful as a reference for this kind of investigation. However, the available permafrost distribution maps in our study (the IPA permafrost map, the CAS permafrost map, and the PZI map) all have very coarse spatial resolutions. Therefore, it is difficult to assess what characteristics of rock glaciers should have that would be suitable for inferring the permafrost distribution based on our present dataset. We decide to keep the original discussion on the lower limit of permafrost in the region, which is a range, and leave the use of individual rock glaciers for permafrost occurrence in the future work.

- There is limited discussion where the authors put their approach and results into context. There are several studies existing which were presenting rock glacier inventories and further information in the neighbouring Kazakh and Kyrgyz Tien Shan. Several ones are in Russian which might be difficult to understand (but may still be considered as they provide valuable information) but there are also several published in English (especially by A. Gorbunov, see references). One more recent paper co-authored by Gorbunov presents also a topographic analysis which would be very interesting to compare your results too.

Authors: We have added an addition discussion section (Lines 409–435) to compare our inventory with the previous studies of rock glaciers in the NTS (Cui and Zhu, 1989; Zhu, 1992a; Zhu et al., 1992b; Liu et al., 1995) and the Kazakh/Kyrgyz Tien Shan (Gorbunov et al., 1992, 1998). The rock glacier studies in the NTS conducted by Cui and Zhu (1989) and Zhu (1992a) reported that most of the identified rock glaciers are tongue-shaped, located at altitudes between 3300 m and 3900 m, and facing north, which are generally consistent with our inventory. The studies of rock glaciers in the Kazakh/Kyrgyz Tien Shan are concentrated in the Djungar Ala Tau, the Ile Alatau, and the Kung öj Ala-Too mountain ranges. The rock glaciers in the Djungar Ala Tau lay lower altitudes comparing to the NTS, probably due to their more northerly latitude. While the altitudinal distributions of the rock glaciers in Alatau and Kung öj Ala-Too are similar with that in the NTS. These previous studies also revealed a homogeneity pattern of the ARGs surface velocities.

Additionally, we found that the topographic analysis conducted by Bolch and Gorbunov (2014) was focusing on the influences of the contributing area parameters (e.g. its area or the headwall height) on the area and minimum altitudes of the rock glaciers. However, such contributing area parameters were not investigated in our inventory. Therefore, it is impractical to compare our results with the topographic analysis of Bolch and Gorbunov (2014).

- Recent findings show a clear seasonal behaviour of the surface velocity (e.g. Wirz et al. 2016). Hence, it is important to mention the acquisition period of the data used to calculate the velocity also in the text. The seasonal effect should also be considered in the discussion.

Authors: The ALOS PALSAR data we used were acquired between 2007 and 2009, and all the data were acquired in summer except for Path 503. Each interferogram spans 46 or 92 days. We state this explicitly in Section 3 (Lines 136 and 139) and Table 1. The seasonal effects may contribute to the spatial heterogeneity of the surface velocities due to the acquisition times for each image pairs are different. We state this point in the revised version in Lines 361–367.

- With respect to the discussion I suggest to include a separate discussion section where you put your approach and results into context of the existing literature. Currently, the discussion section contains only a comparison of the rock glacier derived permafrost estimates to permafrost distribution estimates.

Authors: Following this suggestion, we have added a separated discussion section (Section 5.1) to compare our approach and results with the previous studies of rock glaciers in Tien Shan (Lines 409–435).

- A minor but important point: The Tien Shan is a large mountain range stretching from Uzbekistan into China. Considering the Tien Shan as a whole using "Northern Tien Shan" is not correct. You should rather use "Northeastern Tien Shan" in the title and elsewhere.

Authors: The study area is completely located in China and is termed as "Northern Tien Shan" traditionally. We find very few papers use "Northeastern Tien Shan" in the literature. To avoid confusion, we have changed the "Northern Tien Shan" to "Northern Tien Shan of China" in the title through the paper.

**Specific comments:**

Abstract:

L10: Rock glaciers are widespread not only in western China but in the whole of Tien Shan.

Authors: We have revised the sentences (copied bellow), please see Lines 10–11.

"Rock glaciers are widespread in the Tien Shan Mountain. However, rock glaciers in the Chinese part of the Tien Shan have not been systematically investigated for more than two decades."

L11: There are few recent studies, but your statement is true for Western China. Please correct.

Authors: Done. Please see Line 11.

L24: The approach is interesting for global rock glacier mapping and not only for western China.

Authors: We have rewritten the sentence to "…to map rock glaciers over mountain ranges globally" (Line 22).

1. Introduction:

L28: I am missing the classical monography by Barsch (Barsch 1996) in the references. I'd rather cite Haeberli et al. (2006) instead of 2010 here.

Authors: We have added the reference 'Barsch (1996)' in Line 28 and changed the reference 'Haeberli (2010)' to 'Haeberli et al. (2006)'.

L36: Not sure if a definition of inactive and relict rock glaciers are needed here. This is a cryospheric journal where the readers should know such basic knowledge.

Authors: We removed the definitions of inactive and relict rock glaciers.

L45: Brenning (2005) already mentioned it and used the term intact for active and inactive rock glaciers. You would are able to distinguish which is a major advantage.

Authors: Here we want to express that it is difficult to differentiate active rock glaciers from inactive ones directly through visual interpretation. Thus we chose to keep the original wording, i.e. "active and inactive" (Line 46).

L47ff: You should mention also somewhere that multi-temporal optical images can and were also used to investigate rock glacier velocity (e.g. Gorbunov et al. 1992, by visual interpretation; Kääb et al. 1997 by feature tracking).

Authors: We have added some words to include the use of multi-temporal optical images for investigating the surface velocities of ARGs (Lines 61–63).

L68: Mention here at least one of the existing studies.

Authors: We have added two references (Kenyi and Kaufmann, 2003; Liu et al., 2013) (Line 69).

L74: Include here few of the existing studies of the Kazakh and Kyrgyz Tien Shan.

Authors: We have added several sentences (copied below) (Lines 75–80).

"Rock glaciers are abundant in the high mountains of Central Asia and western China, such as the Tien Shan, Hindu Kush Himalayan, Kunlun Shan and Hengduan Shan. More than 1500 rock glaciers have been inventoried in the Kazakh and Kyrgyz Tien Shan (Gorbunov et al., 1992;1998). Schmid et al. (2015) compiled an inventory with 702 rock glaciers identified from Google Earth images in the vast Hindu Kush Himalayan region. Using these inventories, recent studies have further discussed the rock glaciers-permafrost interactions (Bolch and Gorbunov, 2014; Schmid et al., 2015)."

L80: You may mention here the study by Bolch and Gorbunov for the nearby Northern Tien Shan in Kazakhstan and Kyrgyzstan and maybe also Schmid et al. (2015)

Authors: We have added citations to these papers (Line 80).

2. Study area:

General: Provide more details about the general characteristics of the Tien Shan, the subdivision, and especially the climatic conditions.

Authors: Thanks for your suggestions. We have added detail descriptions of the subdivision and climatic conditions of the Northern Tien Shan of China (Lines 97–99, 111–117).

L97: Be more specific about the annual precipitation amount. The west is wetter than the east but not really wet. The climate is only relative humid for the continental conditions.

Authors: We have added several sentences (copied below) in this section (Lines 111–117).

"There are two long-term meteorological stations in the NTS. The Daxigou meteorological station (86.84 °E, 43.11 °N) in the eastern part is at about 3540 m above sea level. The Snow-cover and Avalanche Research Station (SARS, 84.40 °E, 43.26 °N) lies in a valley in the western part of the NTS, at an elevation of about 1776 m. The mean annual air temperature (MAAT) and the mean annual precipitation around the Daxigou station were about −6 °C and 405 mm in 1990–2008, respectively (Sun et al., 2013). The MAAT near the SARS was about 1.3 ºC, falling to −9.4 ºC at 3580 m. The mean annual precipitation at the SARS was about 830 mm (Shi et al., 2009)."

L102: You cite here almost the same studies than in L75 in the introduction. I suggest to cite in the intro those which provide information about the larger regions or from surrounding

ranges and in the study region section the specific ones.

Authors: Following the suggestion, we have rewritten the relevant sentences in introduction (Lines 75–80) and cited the studies of rock glaciers in the Kazakh and Kyrgyz Tien Shan and the Hindu Kush Himalayan region, such as Bolch and Gorbunov, 2014, Gorbunov et al., 1992, and Schmid et al., 2015.

3.  Methodology:

General: The other reviewers are more experts in SAR processing. Therefore I will not comment on the technical aspects here. However, I'd like to see a better figure where the authors present the identification of moving surface based on the wrapped interferometric phase. Fig. 3 is interesting in this regard but I find it hard to understand how you identified surface displacements based on this image. Maybe a larger image or a zoom would help. Provide a short information about the quality and suitability of the images available at the time of the study in google earth, e.g. had all images good snow conditions and where all of very high resolution?

Authors: We have added two figures (Figure 3c and 3d) for showing the identification of the moving surface based on wrapped interferometric phase. Fig. 3c shows the outline of the rock glacier determined from the variations of phase map, and Fig. 3d shows the corresponding optical image for the rock glacier in Google Earth. Additionally, we selected the Google Earth images that were cloud free and were taken in summer season as the limited extent of snow cover to readily identify ARGs. We state this in the revised version in Lines 164–166.

L148: I agree that active rock glaciers have usually little or no vegetation, but not always (e.g. there are trees on rock glaciers in other parts of the Tien Shan). Hence, write "have usually little or no vegetation" or similar.

Authors: Corrected, please see Line 172.

L151: It is not true that debris-covered glaciers "are usually covered with uniformly thin debris layer". There are manifold studies which show that the debris thickness usually increases towards the terminus and that the surface of a debris-covered glacier is characterised by ice cliffs and supraglacial lakes. It is partly hard to distinguish clearly between debris-covered glaciers and rock glaciers as gradual transitions to moraine-derived rock glaciers exist especially in continental conditions.

Authors: Thanks for pointing out this. We have rewritten the relevant descriptions for the debris-covered glacier (copied below). Please see Lines 174–180.

"We distinguished ARGs from debris-covered glaciers based on the de-correlation conditions of interferograms and their different visual features on the Google Earth images. Compared with rock glaciers, debris-covered glaciers generally move much faster (Janke et al., 2015), which results in large areas of de-correlation in our PALSAR interferograms. The surface of a debris-covered glacier is usually characterized by ice cliffs and supra-glacial lakes (Bolch et al., 2007). And the rooting zones of debris-covered glaciers are continuous with clean glacier ice (Davies et al., 2013; Lukas et al., 2007)."

L161: What is a rooting zone "Z" and where is it in Fig. 1?

Authors: We have enlarged the denotation of the zone "Z" in Fig. 1.

L184f: I find the abbreviations ILP (initial line point) and FLP (front line point) not intuitive as you are mainly interested in the altitude. Humlum (1998) uses RILA (rock glacier initiation line altitude). As you are referring to the maximum and minimum altitude I'd use $h_{min}$ and $h_{max}$ . But you may decide.

Authors: Thanks for your suggestion. We define the ILP and FLP to conveniently determine the parameters of the inventoried ARGs, such as length, aspect, and slope. Therefore, we decide to keep the abbreviations.

L197f: It can be quite difficult to identify the upper boundary. Hence, InSAR is quite promising. Provide more details and examples for how you identified the upper boundary. Ridges and furrows are typical for compressive flow in the lower rock glacier area and, hence, you might have missed parts if you only use these criteria.

Authors: We now added one more example to show the InSAR-determined outline of an ARG, please see Figs. 3c and 3d and Lines 198–200.

L200: Should be Gruber (2012).

Authors: Corrected.

L206: Why do you mention "lower limit of the permafrost distribution" here. You are quantifying the rock glacier parameters.

Authors: We have rewritten the sentence (copied blow), please see Lines 243–244.

"The uncertainties of the FLP and ALP altitudes are determined by the errors of the SRTM digital elevation model, which has a nominal vertical accuracy of less than 16 m (Farr et al., 2004)."

4. Results:

L222f: Move to methods.

Authors: We have moved the sentence to the Method section (Lines 204–205).

L250ff: What is about the influence of the general topography? It could be worth to compare the aspects of the rock glaciers to the aspect distribution of the investigated mountain ranges. The aspect distribution of the rock glaciers in Kazakh and Kyrgyz Northern Tien Shan is clearly influences by the topography.

Authors: We calculated the aspect distribution of the mountain ranges in the study area and found the number of north-facing and south-facing slopes are similar. Therefore, we inferred that the topography in the NTS is not the controlling factoring for the aspect distributions of the MARGs. We presumed that the larger percentage of north-facing MARGs in their quantum than the TARGs is due to aspect distribution of the glaciers. While we noted that the majority of glaciers in the NTS are north-facing.

L252: Kaldybayev et al. (2016) investigate glaciers in Dzhungar Alatau which is close to you study region but is not the Northern Tien Shan.

Authors: Thanks for pointing out this. The reference "Kaldybayev et al., 2016" has been removed in the revised version.

L266: Be careful with the statement about the lower limit. See my general comment above.

Authors: We have added some word to declare the lower limit, please see our replies to the general comments.

L270: What is about the precipitation? I would assume the precip is also of importance.

Authors: Agree. We have rewritten the relevant sentences to state the possible influence of precipitation on the altitude distribution of rock glaciers (Lines 319–322).

L278f: This is an important point and should be discussed more in detail.

Authors: We have further discussed the different altitude distributions of ARGs in the western and eastern parts of the NTS. The paragraph reads now as follows (Lines 323–331):

"Additionally, we found different altitude distributions of ARGs in the western and eastern parts (Fig. 7a). The nearly linear pattern in the east is more apparent than that in the west. We performed a T-test on the similarity of the FLP altitude distributions in the two sub-regions. The P-value is smaller than 0.01 at the 95 % confidence level. Therefore, we conclude that the altitude distribution in the western NTS is significantly different from those located in the eastern part. The relatively scattering pattern in the west indicates that the altitude distributions of the ARGs there are not dominantly controlled by the geographical location

(i.e., longitudes). There could be other factors influencing the altitude distributions in the west. For instance, Bolch and Gorbunov (2014) revealed that the characteristics of the contributing area (e.g., the area, slope, and elevation range) of rock glaciers would influences their altitude distribution as well."

L280ff: You need to consider that the temperature in the blocky material of the rock glaciers can be significant colder than the MAAT (e.g. Gorbunov et al. 2004).

Authors: Thanks for pointing out this. We have rewritten the relevant sentences (copied below) to declare this (Lines 333–336).

"The highest correlation coefficient is found at the factor "MAAT". Although the blocky material of the rock glaciers may lower the ground temperature from the air temperature, this high correction implies that the MAAT may influence ground thermal conditions and thus the formation, evolution, and survival of the ARGs."

L287ff: The first lines of this section describe methods and should be presented in the methods section. In addition, as mentioned information about the time of the year when the velocity was measured are required.

Authors: We move the first sentence into the method section (Line 202). The times of the measured velocities are provided in Table 1.

L302f: The presence of water (e.g. from snow melt or heavy rain fall) has a strong influence on the short term variation. This should be mentioned and discussed along with the acquisition period of the data.

Authors: We now state the influences of precipitations on the derived ARGs surface velocities (Lines 365–367).

5. Discussion

I suggest a separate discussion section where you put your results into context. I suggest to slightly shorten the discussion about the comparison to the existing permafrost maps.

Authors: As suggested, we have shortened the discussion on the comparison of our results to the IPA and CAS permafrost maps.

6. Conclusions

Readers often read the abstract and look at the figures conclusion only before they decide to read the entire paper. I would therefore not use non-common abbreviations in the conclusions and figure captions.

Authors: We have removed the non-common abbreviations in the conclusions and figure captions, such as ARGs and NTS.

L434: Use one decimal digit only. The exact area is quite uncertain.

Authors: We have changed the area of ARGs with one decimal digit.

L442ff: Conclusion 4 is hard to understand.

Authors: Considering the comment of the first referee, we removed the Conclusion 4.

L446: The methods cannot only applied in China.

Authors: We have stated the method can be applied to the other high mountains globally (Line 532).

L450: Reformulate the last sentence. Rock glaciers provide a hint for permafrost occurrence but should be used with care when modelling permafrost distribution.

Authors: We have rewritten the sentence to be rigorous, please see Lines 532–533.

---

## Author Response (AR2)

**Reply to comments by on "Mapping and inventorying active rock glaciers in the Northern Tien Shan of China using satellite SAR interferometry"**

We thank all reviewers and the editor for their comments on our revised manuscript. Only one minor editorial issue was raised. We have fixed it.

Review #1: went through the revised manuscript and spotted only one small typos in the caption of Figure 1 ("... the footprints of seven frames of PALSAR frames ..." should be "... the footprints of the seven PALSAR frames ..."). The authors well considered the reviewer's comments in the revised version of the manuscript.

Authors: Thanks. We have removed the redundant words and it now reads "The black dashed boxes outline the footprints of seven PALSAR frames along Paths 501–507". (2$^{nd}$ line on page 21)